# Multiparametric cellular and spatial organization in cancer tissue lesions with a streamlined pipeline

Multiplex immunostaining analysis remains fragmented, underperforming and labour intensive despite tissue proteomic methodologies achieving ever-increasing marker complexity. Here we propose an open-source, user-guided automated pipeline that streamlines start-to-finish, single-cell resolution analysis of whole-slide tissue, named multiplex-imaging analysis, registration, quantification and overlaying (MARQO). MARQO integrates elastic image registration, iterative nuclear segmentation, unsupervised clustering with mini-batch $k$-means and user-guided cell classification through a graphical interface. We compare and validate MARQO using multiplexed immunohistochemical consecutive staining on a single slide using human tumour and adjacent normal tissue samples. Performance is compared with manually curated pathologist determinations and quantification of multiple markers. We optimize MARQO to analyse diverse tissue sizes from whole tissue, biopsy, and tissue microarray and staining approaches, such as singleplex immunohistochemistry and 20-colour multiplex immunofluorescence, to determine marker co-expression patterns in multiple human solid cancer types. Lastly, we validate CD8+ T cell enrichment in hepatocellular carcinoma responders to neoadjuvant cemiplimab in a phase 2 clinical trial, further showing the ability of MARQO to identify spatially resolved in situ mechanisms by providing multiplex whole-slide single-cell resolution data.

Multiplex immunohistochemistry (IHC) and multiplex immunofluorescence (mIF) are essential imaging tools for determining single-cell-resolution protein co-expression levels while maintaining spatial tissue integrity. Recent technologies, such as co-detection by indexing (CODEX)[1], cyclic immunofluorescence (CyCIF)[2], mIF[3], multiplexed ion beam imaging (MIBI)[4] and multiplexed immunohistochemical consecutive staining on a single slide (MICSSS)[5], have drastically increased the number of targeted proteins stained on one slide, creating many opportunities to better understand tissue organization and cell infiltration. In fact, characterizing and spatially understanding immune cell recruitment and organization in cancer lesions during treatment with immune checkpoint blockade (ICB) treatment has helped elucidate mechanisms of response or resistance and clarify interactions among cells, emphasizing the feasibility and advantage of multiplex imaging technologies[6]. While the experimental workflow of imaging technologies has been increasingly streamlined, the methodology for analysing and producing whole-slide, quantitative multiplex data remains discrepant and computationally intensive, if at all possible, with available third-party analysis tools[7,8]. Although artificial intelligence (AI) is being investigated as a solution, proprietary algorithms make it difficult to accept outputs without confirmation from pathology review. Therefore, a pipeline that reliably streamlines and integrates whole-slide analysis for multiplex and singleplex IHC and immunofluorescence using a largely automated but user-guided approach is critically needed.

✉e-mail: sacha.gnjatic@mssm.edu

In this study, we report the MARQO pipeline and its validation by a pathology team. It uses multiple interchangeable modules including dynamic deconvolution of channels, co-registration, segmentation of cell nuclei and unsupervised clustering of unique cell populations, all performed locally or in a cluster. MARQO leverages parallel and distributed computing to efficiently process workloads, scaling beyond the limitations of a single central processing unit (CPU) by distributing tasks across multiple independent machines in a cluster or cloud environment. This architecture enables rapid analysis of massive cohorts, with on-demand scalability reducing analysis time per cohort to nearly the theoretical minimum (Supplementary Fig. 1). Each step in the pipeline has adjustable parameters that have been optimized and can be adjusted for diverse tissue types and staining protocols (MICSSS[5], COMET immunofluorescence[9], Orion[10] and Vectra[11]). Other recent multiplex analysis tools focus on specific aspects of the quantification methodology, such as co-registration or nuclear segmentation (ASHLAR[12] and MCMICRO[8]), requiring the user to toggle across applications and learn new software. This fractionation of software is a hurdle for harmonized quantification across platforms, an essential requirement of large networks for correlative science, such as CIMAC-CIDC[13]. MARQO uses the iterative qualities of multiplex data to help refine repetitive data, using the multiplex aspect of the assay to its advantage. Moreover, whereas existing tools are mostly released as GitHub repositories, only operational through the Command Prompt interface, MARQO is easily operational for both computational and non-computational users via a Command Prompt interface and a graphical user interface (GUI). Lastly, we have used a user-guided approach to characterize cell populations because fully automated decisions about positive signals are not yet clinically acceptable without pathologist confirmation. The approach of unsupervised clustering followed by a supervised binarization of each cluster by the user permits MARQO to accurately quantify and cluster cell populations across multiple platforms, tissue types and markers of interest, without the need to train the machine learning algorithms, while still permitting closely monitored quality control by the user.

We applied MARQO to whole-slide MICSSS data, a technology developed and broadly used by our laboratory[14–16]. MICSSS data remain difficult and labour intensive to analyse with existing tissue analysis tools, such as HALO[17] and Visiopharm[18]. Moreover, image-processing software, such as QuPath[19], still lacks key functionalities for steps in multiplex processing that we have prioritized in MARQO, including the ability to elastically register whole-slide images, segment cells iteratively across markers to produce one composite segmentation mask and cluster cell populations via unsupervised learning. Therefore, MARQO provided the necessary framework to study spatial immune responses to ICB in cancer. Although neoadjuvant ICB targeting the PD1/PDL1 axis has revolutionized treatment approaches for various cancer types[20], the response to immunotherapy varies in success[21] and many patients experience no clinical benefit. CD8 T cell enrichment has been associated with the response to ICB[22–24]. Leveraging a trial of neoadjuvant cemiplimab (anti-PD1 antibody; ClinicalTrials.gov NCT03916627, cohort B) in patients with hepatocellular carcinoma (HCC)[25], we recently corroborated these findings in responders to ICB. However, to fully understand the spatial dynamics and mechanisms driving the response to treatment, we assessed multiple cell populations using MICSSS and used MARQO to resolve the multiparametric cellular and spatial data in responders to ICB relative to non-responders. With MARQO, we detected CD8 T cell enrichment in specific tissue areas defined by a pathologist as tumour, fibrosis and necrosis. In addition, we analysed immune cell organization and proximities in tumours pre- and post-treatment. As a result of this method of multiplex imaging and analysis, we are better able to elucidate the quantitative mechanisms that underlie the responses and resistance to neoadjuvant anti-PD1 treatment.

## Results

### MARQO enables analysis of diverse imaging technologies
MARQO was tested and validated for its ability to analyse MICSSS, mIF (COMET IF) and singleplex IHC (multiple chromogens) using the workflow described in Fig. 1. Following each respective protocol for producing and scanning all stained tissue slides, MARQO can analyse these datasets locally or via a cluster. To begin the analysis, the user operates the Command Prompt or downloadable GUI to quality control each sample by specifying input data types, including the type of assay and the name of each stain (Supplementary Video 1). This is followed by a tissue masking and, if applicable, a preliminary registration step, to ensure that the tissue is being correctly captured. The next steps of the pipeline run automatically and depend on the type of data being analysed. For example, while IHC technologies require a colour deconvolution to separate the chromogen from the nuclear counterstain (usually haematoxylin), multiplex immunofluorescence images contain a single stain per channel in a multidimensional array, which usually requires a denoising step. The pipeline splits the tissue into evenly sized tiles and, because of its advanced workload distribution architecture, enables parallel analysis of these tiles across multiple independent CPUs or machines. This substantially accelerates computation, allowing for the analysis of hundreds to thousands of tiles simultaneously, constrained only by the user's available hardware resources. During the per-tile analysis, the registration, segmentation and quantification modules are enacted, again dependent on the technology being analysed. While some mIF technologies require the registration step[2], others are already aligned and can forgo this step[9]. This step is followed by an unsupervised clustering module, in which each stain is treated as an independent input, and segmented cells are compartmented into multiple clusters. The user is subsequently prompted to perform another quality control to review the performance of the analysis and binarize the positivity of each cluster per marker.

To promote seamless integration into workflows and other software, MARQO produces intermediate file types compatible with other platforms, such as QuPath, that the user can conveniently drag and drop for continued analyses (Supplementary Table 1).

### Improved segmentation via iterative cell staining
For MICSSS and some mIF, repetitive nuclear staining can be harnessed to clarify cell boundaries and refine imaging data. Whereas some mIF technologies use 4′,6-diamidino-2-phenylindole (DAPI) for nuclear counterstaining, MICSSS counterstains nuclei with haematoxylin per staining cycle iteration. For this reason, we stress the benefits of multiple nuclear or cytoplasmic stains while using MICSSS in our validation. MARQO performs a new registration technique for MICSSS in which each tile per iterative stain is matched to the respective tile of the first stain that encompasses roughly the same tissue area. This process results in parallel batches of tiles that are aligned by their deconvoluted nuclear stains, which is assumed to remain consistent across each batch because there is no chromogen or other-coloured artefact in this channel (Supplementary Fig. 2). MARQO then performs nuclear segmentation iteratively across stains per batch of tiles via an open-source pretrained package named StarDist[26] (Fig. 2a). The pipeline leverages the strength of multiplex nuclear staining to enhance segmentation accuracy. MARQO systematically analyses each nuclear object identified across multiple stains. A nuclear object is retained in the final composite segmentation mask if its centroid is consistently detected in at least 60% of the iterations (default threshold) within a predefined distance of 3 μm (default, equivalent to 30% of the average immune cell diameter of 10 μm). This threshold accounts for potential registration errors, with both parameters fully adjustable for specific needs. Reconciling across multiple stains permits the pipeline to have a larger sample size to evaluate whether a cell is a true-positive-segmented cell versus a red blood cell (RBC), an artefact or a cell lost from tissue damage that was deemed a false-positive-segmented cell. Similarly, on the

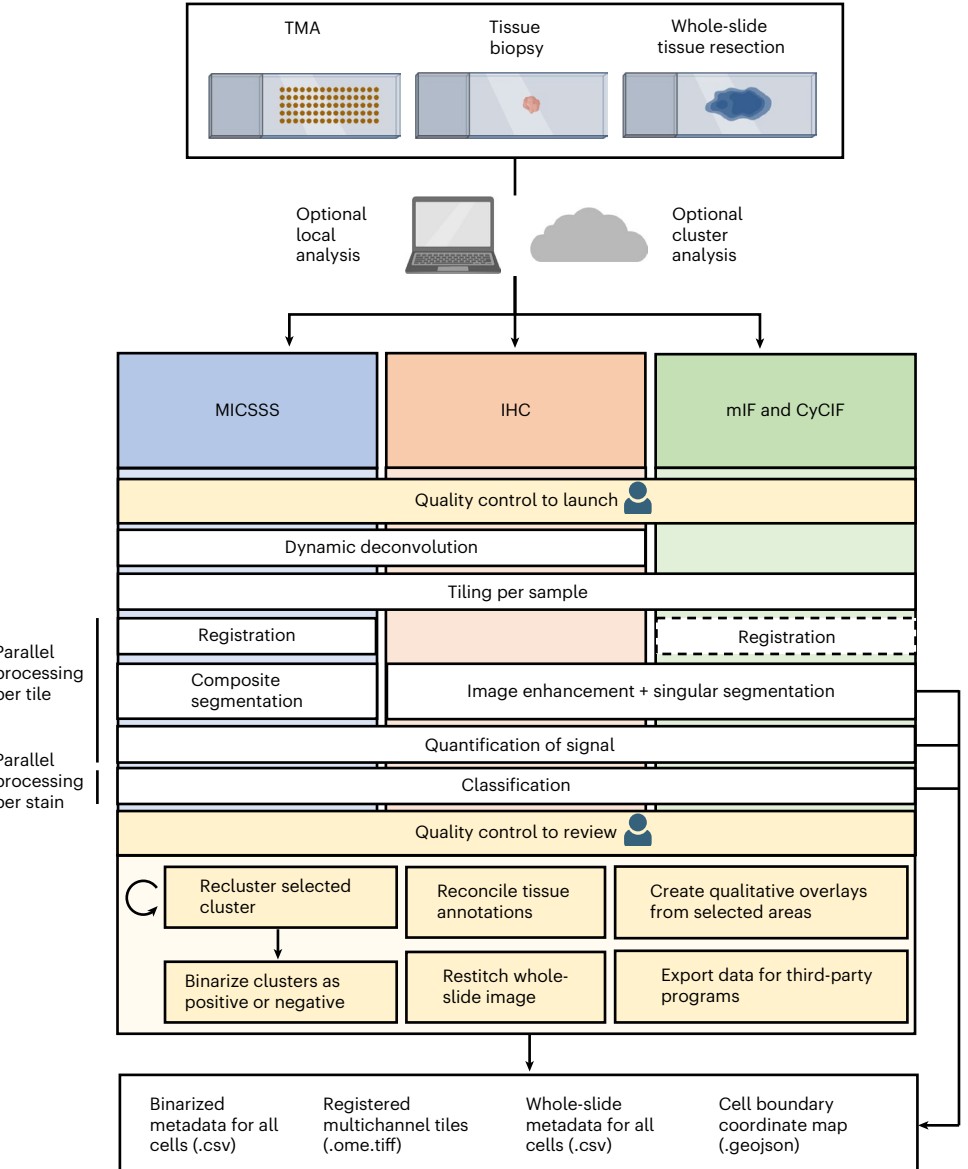

**Fig. 1 | MARQO pipeline overview.** Variable formalin-fixed and paraffin-embedded (FFPE) tissue sizes from small TMA cores to whole-slide tissue resections can be analysed using a tiling architecture, which can be performed locally or externally via a cluster. The pipeline is compatible for slides stained with MICSSS, singleplex IHC, or mIF and CyCIF. After the user provides a brief quality control when importing data, the pipeline independently processes each staining technology differently. MICSSS and IHC undergo dynamic deconvolution to extract nuclear and chromogen staining. All technologies are converted into numerous smaller, overlapping tiles for faster, parallel processing. The MICSSS and CyCIF technologies undergo a series of registrations. All technologies are segmented for cell nuclei, where MICSSS uses a composite methodology from its iterative nuclear staining.

All technologies quantify the chromogenic positivity staining, nuclear staining and morphological information. These metadata are used for cluster-based classification, after which the user quality controls to assess which markers the clusters truly stain positively or negatively for. The user can use MARQO downstream analysis tools to further review data. These modules can reconcile tissue compartment annotations from QuPath, recreate large chromogenic or whole-channel image files that combine multiple individual tiles, create figure-ready qualitative overlays to visualize cell populations and export data for third-party programs. At multiple checkpoints in the pipeline, intermediary files that are also easily integrated into third-party programs are exported. Illustrations created with BioRender.com.

basis of staining and available metadata, the pipeline further refines the mask by eliminating hypothesized RBCs, which we have validated to the manual work of a pathologist (Supplementary Fig. 3a–c).

To assess and validate the performance of MARQO's composite segmentation, we used samples stained with an onco-immune-targeted MICSSS panel. We analysed a cohort of patients with HCC who received neoadjuvant cemiplimab treatment followed by surgical resection; for these patients, we analysed pretreatment biopsies and HCC resections. Using ten 500 μm × 500 μm regions of interest (ROIs) from these tissues, we compared the composite segmentation performance of MARQO with that of manual segmentation by a pathologist using a conventional third-party analysis tool, QuPath[19]. We found that MARQO's segmentation performed well across heterogeneous densities, which is usually problematic because cell segmentation is set up for global and homogeneous density (Fig. 2b). Overall, MARQO segmented ±10% of total cells determined manually in >50% of all

ROIs and ±20% of total cells determined manually in >90% of all ROIs (Fig. 2c,d). MARQO achieved an average Sørensen–Dice coefficient (Dice score) of 83% using a point-in-polygon strategy, which considered a MARQO-segmented cell a true cell if the pathologist had clicked on a point within the cell's nuclear boundaries (Supplementary Fig. 3d). In accordance with recent guidelines for including tissue variability[27] and pathologist supervision[28] for improving automated pathology algorithms, these data support the claim that MARQO's composite segmentation algorithm more accurately segments cells compared with the more conventionally used singular stain segmentation method, and it segments similarly to the manual segmentation performed by a pathologist. In summary, we believe MARQO's composite segmentation and elastic registration methodology provide the necessary multiplex preprocessing to ensure the most accurate cell quantification and downstream analysis (Supplementary Fig. 4).

## Fast unsupervised clustering improves cell classification

Once a batch of tiles representing an entire tissue section is analysed, a features table of segmented cells is appended with metadata associated with all segmented cells within that region, which provides the framework for the classification module. Although many current classification approaches leverage a supervised machine learning algorithm trained per dataset[29], we envisioned an all-encompassing model that could be applied across multiple technologies, tissue types and markers without additional training. Moreover, our pipeline contains tunable parameters that, despite being optimized for the technologies we present here, can be tailored for other technologies. These values are all documented with descriptions in a markdown section on our application, allowing the user to adapt them as necessary. The defaulted values were optimized for the technologies presented here within reasonable computation times; therefore, we recommend these values be altered only with an adequate understanding of their use or experience in computational image analysis. We also wanted to permit close, user-guided interpretation of the results because learning models are not mature enough to be used entirely without human supervision in digital pathology[28]. MARQO combines unsupervised clustering for all cells and supervised classification per marker for each cluster by the user. MARQO performs a faster clustering approach for larger samples (mini-batch $k$-means) and a more intensive, but lengthier, clustering approach for smaller samples (Gaussian mixture model), if needed. By default, and for the analyses in this article, MARQO uses the mini-batch $k$-means algorithm because of its scalability and quick performance for larger datasets without significant compromise in accuracy (Supplementary Fig. 1c). When the automated clustering finishes, the user operates a GUI to assess which of the produced clusters of cells are positive or negative per marker (Fig. 3a and Supplementary Video 2). If the user deems a certain cluster still mixed with positive and negative cells, then the user may 'reclassify' a cluster to output a new round of subtiers within the original tier, permitting the user to closely fine-tune classification outcomes.

To validate the performance of our classification module, we leveraged our cohort of resected HCC samples and identified four markers that have unique challenges when quantifying with currently available platforms: nuclear marker FOXP3 (regulatory T ($T_{reg}$) cells), circular shape membrane marker CD3 (T cells), asymmetrical shape membrane marker CD68 (macrophages) and cytoplasmic marker PanCK (tumour cells; Fig. 3b). We compared MARQO-positive classification results after user quality control with the number of positive cells manually counted by a pathologist. We used 34 ROIs from diverse samples and tissue types, which were also deemed as good- or poor-quality areas by the pathologist based on the degree of tissue damage and staining artefacts. Using the manually counted cells as the predictive model, MARQO achieved 83% specificity and a Spearman correlation coefficient $r = 0.95$ ($P < 0.0001$) for CD3 and 90% specificity and $r = 0.90$ ($P = 0.0002$) for FOXP3 (Fig. 3c–e and Supplementary Fig. 5). It also achieved 97% specificity and $r = 0.70$ ($P = 0.2333$) for CD68 and 85% specificity and $r = 0.60$ ($P = 0.4167$) for PanCK, markers that are conventionally difficult to quantify due to their amoeboid-like shapes and heterogeneous staining. We also visualized the performance of MARQO classification for additional markers, including PD1, CD8, Ki67, αSMA, CD20 and MZB1 (Supplementary Fig. 6). These data support the claim that MARQO's fast user-guided classification performs similarly to the lengthy, manual annotations performed by a pathologist, which grants the user time and the ability for continued downstream analysis.

## Diverse tissue sizes and types are processed using MARQO

MARQO can theoretically process tissue of any size due to its tiling architecture. However, during multiplex staining, different-sized tissue sections possess unique biological, technical and computational challenges: smaller tissues have increased relative edge effect staining[30] and may incidentally contain rare, niche cell populations, which are less likely to be isolated during unsupervised clustering; meanwhile, larger tissues are more prone to tissue damage[31], frequently suffer from tissue folding and require vast computational resources. To assess whether MARQO performs similarly across tissues of different sizes, we analysed tissue microarray (TMA) cores, core needle biopsies and samples from surgical resections that were processed with MICSSS (Fig. 4a). Moreover, we assessed and validated classification consistency across diverse solid tumour types, as diverse tissues can contain unique morphologies and marker expression patterns[32] (Fig. 4b). Following MARQO analysis and user review quality control, we determined marker densities for CD3 and PanCK. We chose these two markers due to their broad use in imaging as T cell and tumour markers. Moreover, CD3 is a membrane marker for smaller cells, while PanCK is a cytoplasmic marker for larger cells, enhancing validation through diverse cell testing. The cohort included whole-slide resected tissue of non-small-cell lung cancer (NSCLC) and TMA cores from the following tissue types: NSCLC, head and neck squamous cell carcinoma, colorectal cancer, breast cancer, epithelial ovarian cancer, pancreatic ductal adenocarcinoma, glioblastoma, renal cell carcinoma and melanoma. We compared densities generated by MARQO or by a pathologist through a standardized methodology on QuPath[19]. We determined that CD3 had an overall Spearman $r = 0.98$ ($P < 0.0001$)

**Fig. 2 | Detailed singular and composite segmentation. a**, An example batch of tiles that spans 1–$n$ stains from MICSSS undergoes new composite segmentation. The first column shows the original staining before segmentation, then the nuclear segmentation mask if each marker were independently segmented, then the nuclear segmentation from reconciling multiple nuclear stains, followed by the composite segmentation with nuclear boundaries extended by 3 pixels to simulate cytoplasm. The last column provides examples of the improvements that composite segmentation has over the singular segmentation. Scale bar, 30 μm. **b**, An example tile of MICSSS depicts MARQO composite segmentation across diverse cell densities; the zoomed-in images show how MARQO similarly segments in low-, medium- and high-density areas with a 1.6× linear magnification relative to the main image. Scale bar, 50 μm. **c**, Example tiles of MARQO-segmented cells compared with a pathologist's manual identification of true cells for low-density (left) and middle- to high-density (right) regions. If a manually clicked point resides within the nuclear boundaries of a cell segmented by MARQO, that cell is considered 'MARQO+, manual+' (red). If no point resides within the nuclear boundaries of a cell segmented by MARQO, that cell is considered 'MARQO+, manual−' (blue). If a clicked point did not reside within any MARQO-segmented cell nuclear boundaries, an overlaid circle with a radius of 3 pixels (green) is plotted, simulating a 'MARQO−, manual+' cell. Scale bars, 20 μm. **d**, Total cell counts segmented are plotted for all tiles ($n = 15$) for our manual quantification (left) versus our MARQO automated quantification (right).

and PanCK had an overall $r = 0.90$ ($P < 0.0001$) across all tested tissues (Fig. 4c). These findings suggest that MARQO performs similarly to the manual quantification by pathologists for diverse tissue sizes and types.

## MARQO integrates workflow for multiple staining technologies

Next, we assessed MARQO's performance to analyse diverse types of assay. While MARQO has been optimized for quantifying cell subset

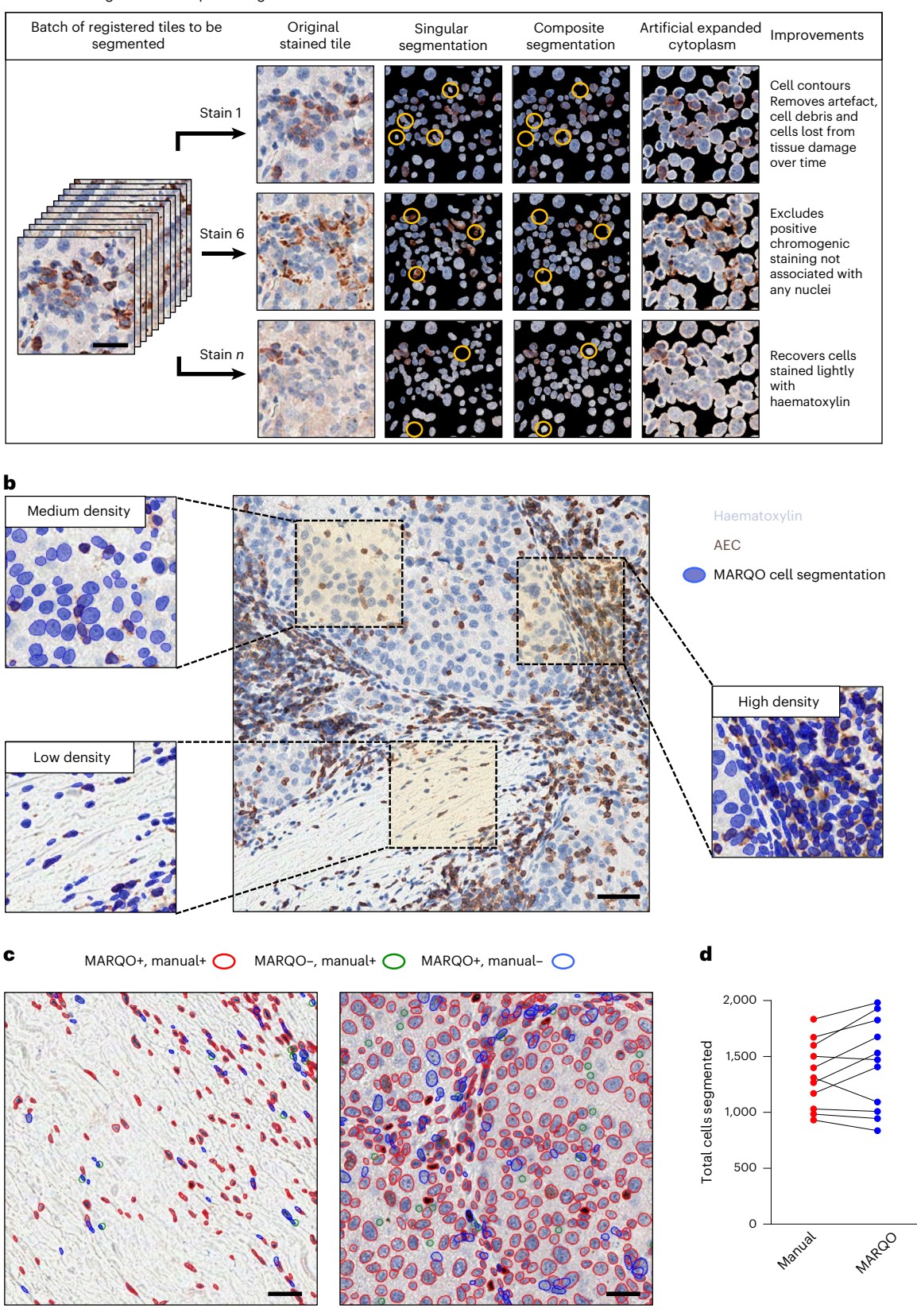

**a** Detailed singular and composite segmentation

**b**

**c** MARQO+, manual+ ◯   MARQO–, manual+ ◯   MARQO+, manual– ◯

**d**

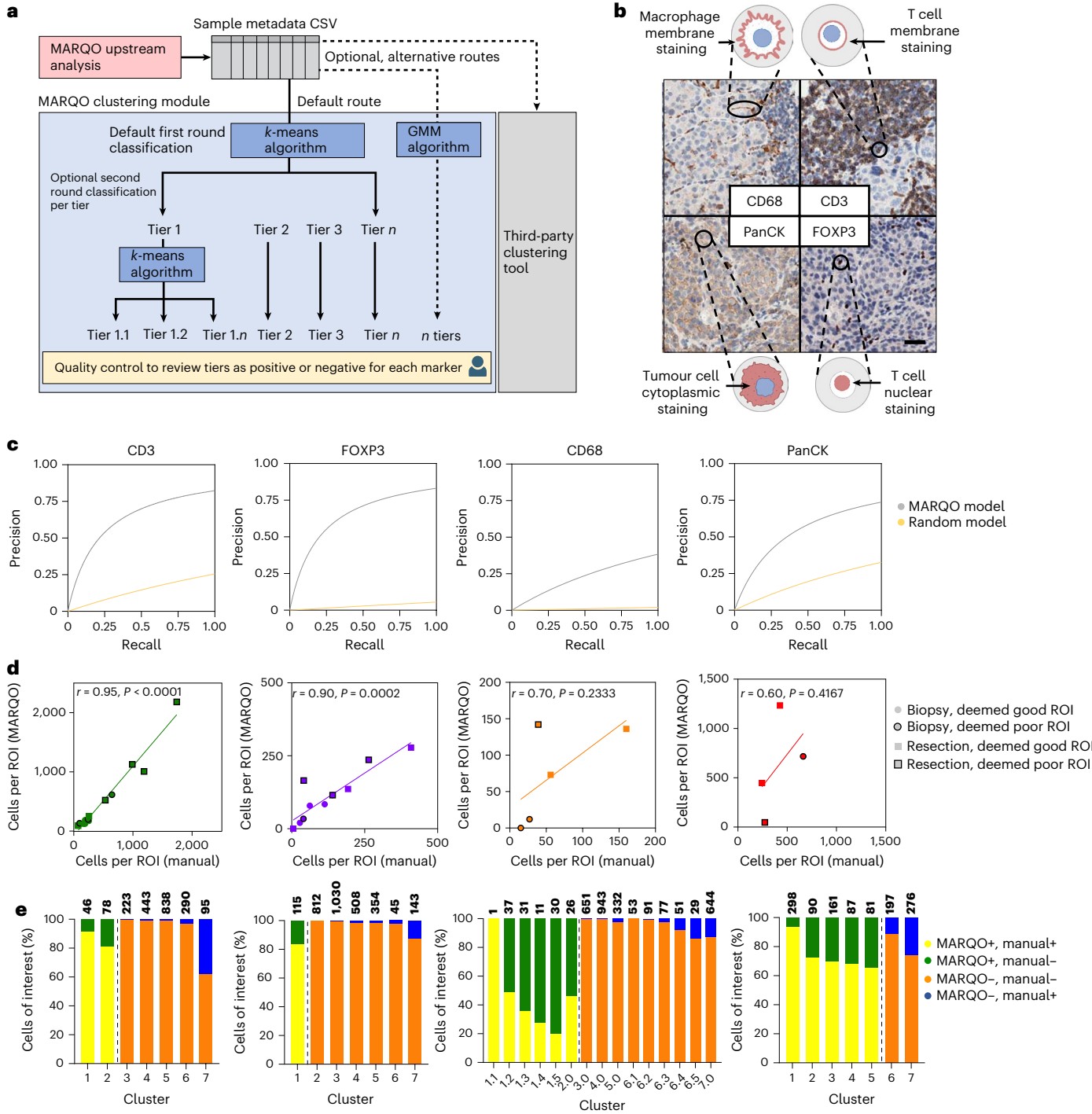

**Fig. 3 | MARQO classification with validation. a,** Following upstream analysis and the generation of a metadata CSV for each sample, MARQO classifies cells using either the default *k*-means algorithm or alternative methods, such as GMM clustering or third-party tools. MARQO categorizes cells into a predefined number of clusters, which users can inspect interactively via the MARQO application. Users may further subdivide clusters into subclusters by rapidly applying *k*-means clustering repeatedly to the selected cluster, enabling detailed user-guided inspection of each cluster or subcluster. **b,** We validated with four distinct markers: FOXP3, a nuclear marker; CD3, a T cell membrane marker with a circular shape; CD68, a macrophage membrane marker with an asymmetrical shape; and PanCK, a cytoplasmic tumour marker. Scale bar, 50 µm. **c,** Precision versus recall curves are provided for markers CD3, FOXP3, CD68 and PanCK, comparing the MARQO classification performance with manual positivity annotations conducted by a pathologist (predictive model). A random model is included as a baseline, representing theoretical performance from random label assignment based on uniform distribution. **d,** Scatter plots

with regression lines illustrate correlations between the total number of cells classified as positive manually by a pathologist versus MARQO across 34 distinct ROIs for the 4 selected markers. Each point represents either biopsy or resection tissue, categorized as having good- or poor-quality staining by the pathologist. ROIs were chosen to represent varied staining quality and regions with tissue damage. Non-parametric Spearman's correlation coefficients (*r* values) and linear regression analyses were used for markers CD3 (*P* < 0.0001), FOXP3 (*P* = 0.0002), CD68 (*P* = 0.2333) and PanCK (*P* = 0.4167). **e,** Stacked bar graphs depict a representative ROI for each selected marker, illustrating the user-based cluster classification per sample and comparing it directly with manual annotations provided by the pathologist for the same ROI. Colours within bars indicate the proportions of cells classified as positive or negative. Dashed lines separate clusters selected as positive (left) or negative (right) by MARQO. Total cell counts per cluster are annotated above each bar. CD68 exemplifies user-driven reclassification into subclusters. Corresponding ROIs are available in Supplementary Fig. 5.

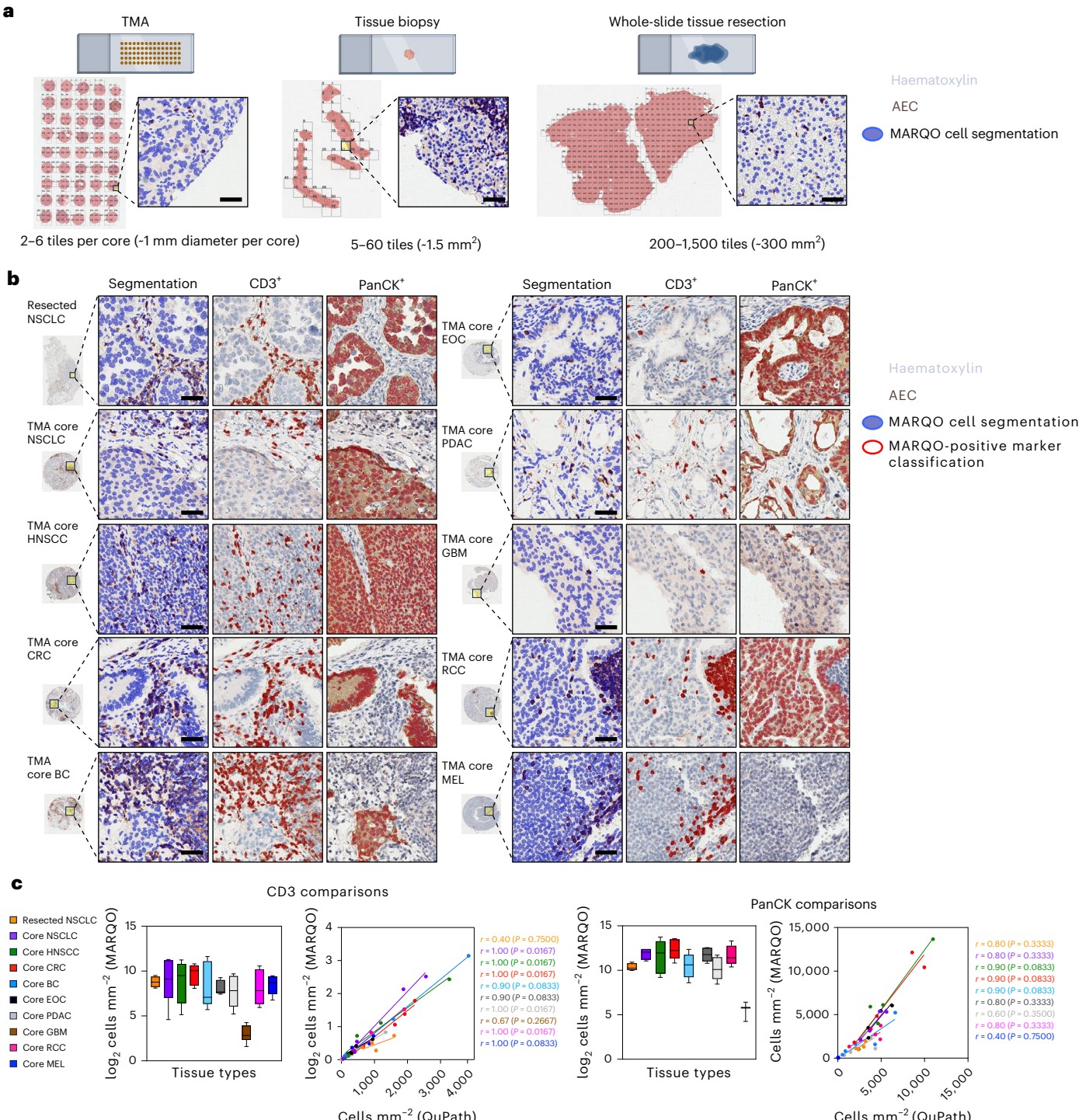

**Fig. 4 | Validation across diverse tissue sizes and types. a**, We tested small TMA cores, tissue biopsies and whole-slide tissue resections on MARQO, with these tissues tiled and segmented here. Scale bars, 50 μm. **b**, MARQO's segmentation and classification performance after quality control by the user for markers CD3 and PanCK are shown for the following tissue sizes and types: whole-slide resected tissue NSCLC and TMA cores for NSCLC, head and neck squamous cell carcinoma (HNSCC), colorectal cancer (CRC), breast cancer (BC), epithelial ovarian cancer (EOC), pancreatic ductal adenocarcinoma (PDAC), glioblastoma (GBM), renal cell carcinoma (RCC) and melanoma (MEL). Scale bars, 50 μm.

**c**, Box plots depicting MARQO densities and scatter plots with regression analysis lines comparing MARQO densities with QuPath-derived densities for the corresponding tissue sizes and types are shown here for markers CD3 and PanCK ($n = 4$ per tissue type). Box plots show the median (centre line), 25th and 75th percentiles (bounds of the box), and minimum and maximum values (whiskers). For the scatter plot, each plotted point corresponds to a biologically replicated resection sample or TMA core. Non-parametric Spearman's correlation $r$ values and linear regression analyses were used and associated $P$ values are shown per tissue type per regression.

density based on consecutive stains and multiple markers, we assessed its ability to analyse singleplex IHC, an assay often used for routine staining of markers of interest. We compared marker densities for CD3-stained cells with another commonly used detection reagent,

chromogen diaminobenzidine (DAB), and used the same reagent used for MICSSS, chromogen 3-amino-9-ethylcarbazole (AEC), to look at PDL1, a marker used in the clinic for diagnostic and prognostic purposes[33]. While we acknowledge that this validation is for research

purposes only, we show that with further clinical validation MARQO may be used in a clinical setting. We determined densities acquired by a pathologist with QuPath on an independent cohort of five samples. For CD3, we determined a Spearman $r = 0.60$ ($P = 0.3500$) and for PDL1 we determined $r = 1.00$ ($P = 0.0167$) for positively classified cells between MARQO and QuPath (Fig. 5a). Similarly, to confirm the accuracy of MARQO for segmenting and characterizing cell populations for non-IHC multiplex technologies, we analysed four whole-slide NSCLC samples via COMET immunofluorescence, a technique using mIF. We compared cell densities of FOXP3, CD3, CD68 and PanCK between MARQO and QuPath, obtaining an overall $r = 0.92$ ($P < 0.0001$) for all the markers (Fig. 5b). In addition to testing and validating singleplex IHC and COMET immunofluorescence, we also tested other platforms on MARQO, including Orion[10], CODEX[1] and CyCIF[2] (Supplementary Fig. 7 and Supplementary Table 1). Consistent with our previous results, these data suggest that MARQO retains high accuracy when analysing diverse tissue and assay inputs.

### MARQO detects CD8 T cell enrichment in therapy responders

Multiple downstream analyses can be performed in the MARQO GUI from the final summary table, which can facilitate further quantifications and analyses (Supplemental Video 2). We leveraged a clinical trial cohort of 18 patients with early stage HCC treated with neoadjuvant cemiplimab and used MARQO to better understand the immune cell infiltration in responders ($n = 6$) versus non-responders ($n = 12$). We used MICSSS and MARQO to analyse pretreatment HCC biopsies (baseline) and HCC surgical resections after two doses of cemiplimab (post-treatment) and observed that not only did the response to cemiplimab modify the global immune infiltration compared with non-responders but the immune landscape was already different at baseline between these patients (Fig. 6a). A pathologist demarcated tumour, non-tumour-adjacent liver (adjacent), fibrotic and necrotic regions in each tissue section using QuPath to understand the specific cell distribution among the samples (Fig. 6b,c). We found that, unsurprisingly, non-responders were mostly represented by tumour tissue, whereas this compartment completely disappeared in responders post-treatment. Specifically, we were interested in CD8 T cells, as we previously characterized their clinical relevance and spatial interactions in response to cemiplimab treatment[34]. Using our GUI, we reconciled the QuPath annotations with cell classification, producing a table that identified the location of each cell within tumour, adjacent, fibrotic and necrotic areas. We detected enrichment of $CD8^+CD3^+$ T cells in fibrotic and necrotic areas of post-treatment responders compared with baseline ($P = 0.3398$ and $P = 0.0267$, respectively; Fig. 6d). Interestingly, we found that CD8 T cells were already enriched in responders at baseline compared with non-responders ($P = 0.008$). To further explore this phenomenon, we used a neighbourhood analysis approach to determine the shortest distance between cell types of interest per cell and how the distance varied with treatment in the different annotated areas. We found that CD8 T cells tend to be very close to each other in responders compared with non-responders in fibrotic and necrotic regions both at baseline ($P = 0.0012$ and $P = 0.0187$, respectively) and in post-treatment fibrosis ($P = 0.0162$), suggesting immune aggregations (Fig. 6e). In comparison, CD8 T cells were generally found within 50 µm of B cells and $T_{reg}$ cells, suggesting a potential interaction (Supplementary

Fig. 8a). From the B cell perspective, we observed that they were close to each other in responders compared with non-responders in post-treatment necrotic regions ($P = 0.0291$), and close to CD8 T cells in responders' fibrotic regions in post-treatment compared with baseline ($P = 0.0004$) (Supplementary Fig. 8b). In summary, these results show the quantitative metrics available using MARQO and provide the first example of cellular and spatial reconciliation of cancer lesions using MICSSS.

## Discussion

In this study, we introduced and validated the MARQO pipeline to quantitatively analyse whole-slide singleplex IHC, multiplex IHC and mIF, enabling quantitative interpretation of cellular and spatial organization in cancer tissue lesions. Through its diverse modules and adjustable parameters, MARQO can be tailored to analyse data from multiple platforms, tissue types and markers of interest, outputting comprehensive co-expression whole-slide data using a single platform. MARQO and its intermediate diagnostic files seamlessly integrate with third-party downstream analysis tools. Moreover, MARQO provides a GUI-based module that formats the summary outputs to the software of interest. Specifically, MARQO provides a throughput whole-slide analysis tool for MICSSS, a task that has so far been suboptimal or labour intensive using standard tools such as QuPath[19], Halo[17] or Visiopharm[18] due to the iterative nature of the staining protocol and need for precise image alignment.

AI approaches hold immense promise for analysing diverse tissue types. However, their implementation often demands extensive training datasets and may lack robust mechanisms to ensure signal specificity. To address these limitations while maintaining high specificity, we developed a user-guided method that uses dynamic thresholds and manual selection of positive versus negative signals to cluster and classify cell populations effectively. This method offers adaptability and precision while paving the way for future integration of AI-based methodologies to enhance specificity validation and dataset generalization. Although this approach is designed to be user friendly for non-experts, it requires pathology expertise and supervision at this step, similar to other platforms dependent on tissue organization knowledge. To alleviate the workflow, we enable users to perform these steps either per sample or across groups of selected samples. In addition, we provide detailed descriptions of all sensitivity parameters, allowing users to tailor the pipeline to their specific needs and applications.

MARQO's performance, benchmarked against pathologist annotations on the samples presented here, highlights its effectiveness but also underscores the importance of user quality control during this transitional phase of digital pathology. Technical challenges, such as tissue damage and border artefacts, can impact automated performance, further emphasizing the need for expert oversight. In high-throughput scenarios or particularly complex cases, unbiased automated methods have the potential to complement or even surpass manual annotations by reducing human variability[35]. As large model training continues to advance, these user-guided workflows may evolve into fully autonomous AI-driven decision-making tools. Nonetheless, with the current user-guided approach, we are confident that MARQO can provide robust and reliable data for a wide range of preclinical and clinical tissues across various pathologies.

**Fig. 5 | Integrative platform for other staining technologies. a**, Singleplex IHC examples segmented and classified after quality control by the user for marker CD3 with chromogen DAB (top) and marker PDL1 with chromogen AEC (bottom). Scatter plots with non-parametric Spearman regression analysis compare MARQO densities with QuPath-derived densities for CD3 (top) and PDL1 (bottom). Spearman correlation $r$ values with associated $P$ values are shown per marker. Scale bars, 30 µm. **b**, mIF using COMET immunofluorescence technology to assess MARQO's ability to segment and classify another multiplex technology after user quality control. We depict a representative tile with cellular nuclear

stain DAPI and the 4 previous overlapping markers of interest ($n = 4$ per marker): FOXP3, CD3, CD68 and PanCK. Stemming from this tile (right) is the single DAPI stain, adjacent to the MARQO segmentation mask overlaid on the DAPI image. Also stemming from the tile (bottom) is each of the marker channels of interest above that marker channel with MARQO's classification channel after user quality control overlaid. The top-right panel is a scatter plot with non-parametric Spearman regression analysis to compare MARQO densities with QuPath-derived densities for all four markers. Spearman correlation $r$ values with associated $P$ values are shown per marker. Scale bar, 30 µm.

An inherent limitation of whole-slide IHC and immunofluorescence analysis is the size of the required computational resources. While we and other analysis tools[8,12] have used a tiling algorithm to parallelize processing and reduce computation time, multiple CPUs are needed, which are not always available with local resources. Considering this, biopsy and TMA analyses are feasible locally, but we strongly recommend cluster resources for whole-slide multiplex imaging analysis. Another limitation includes the quantification of individual

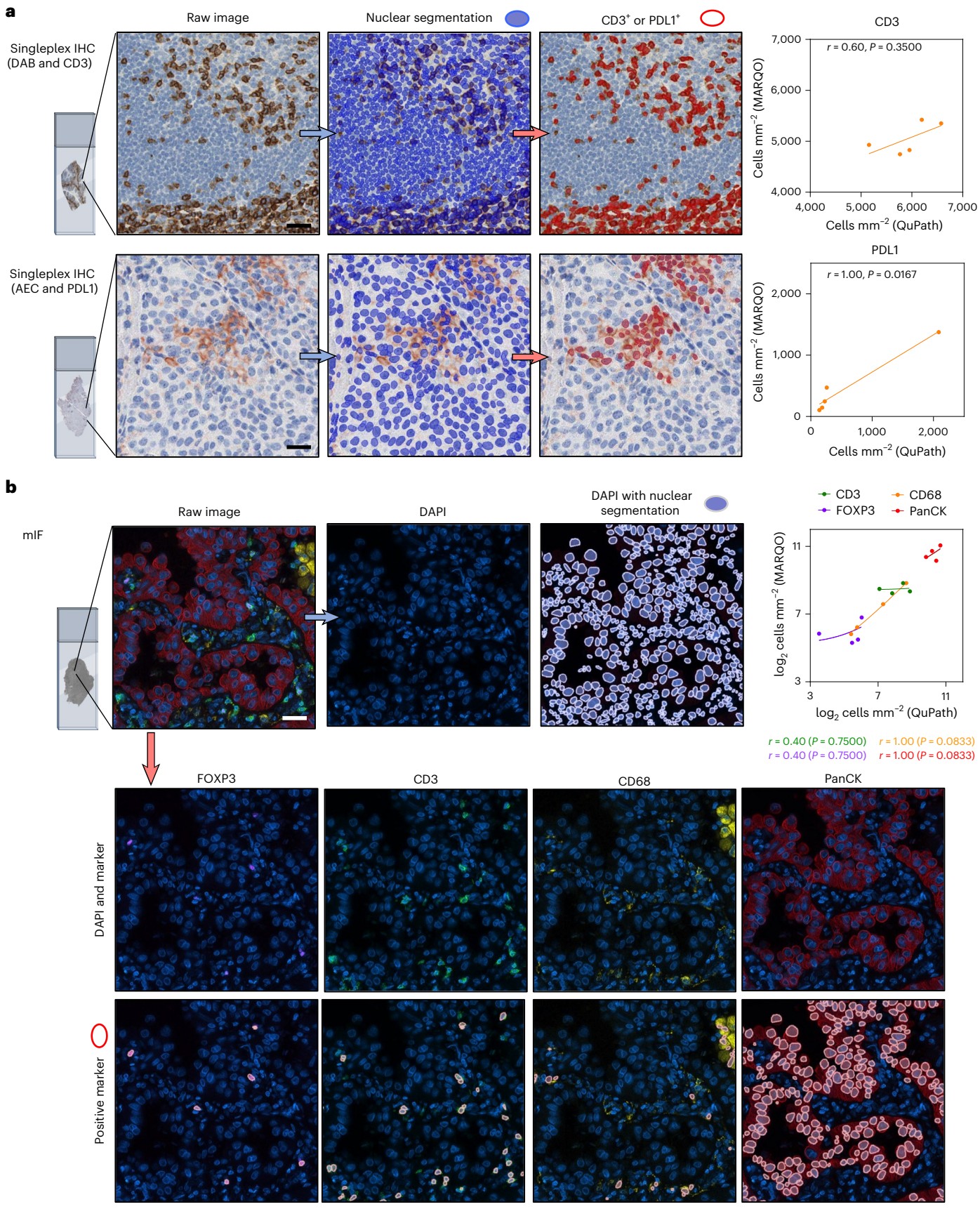

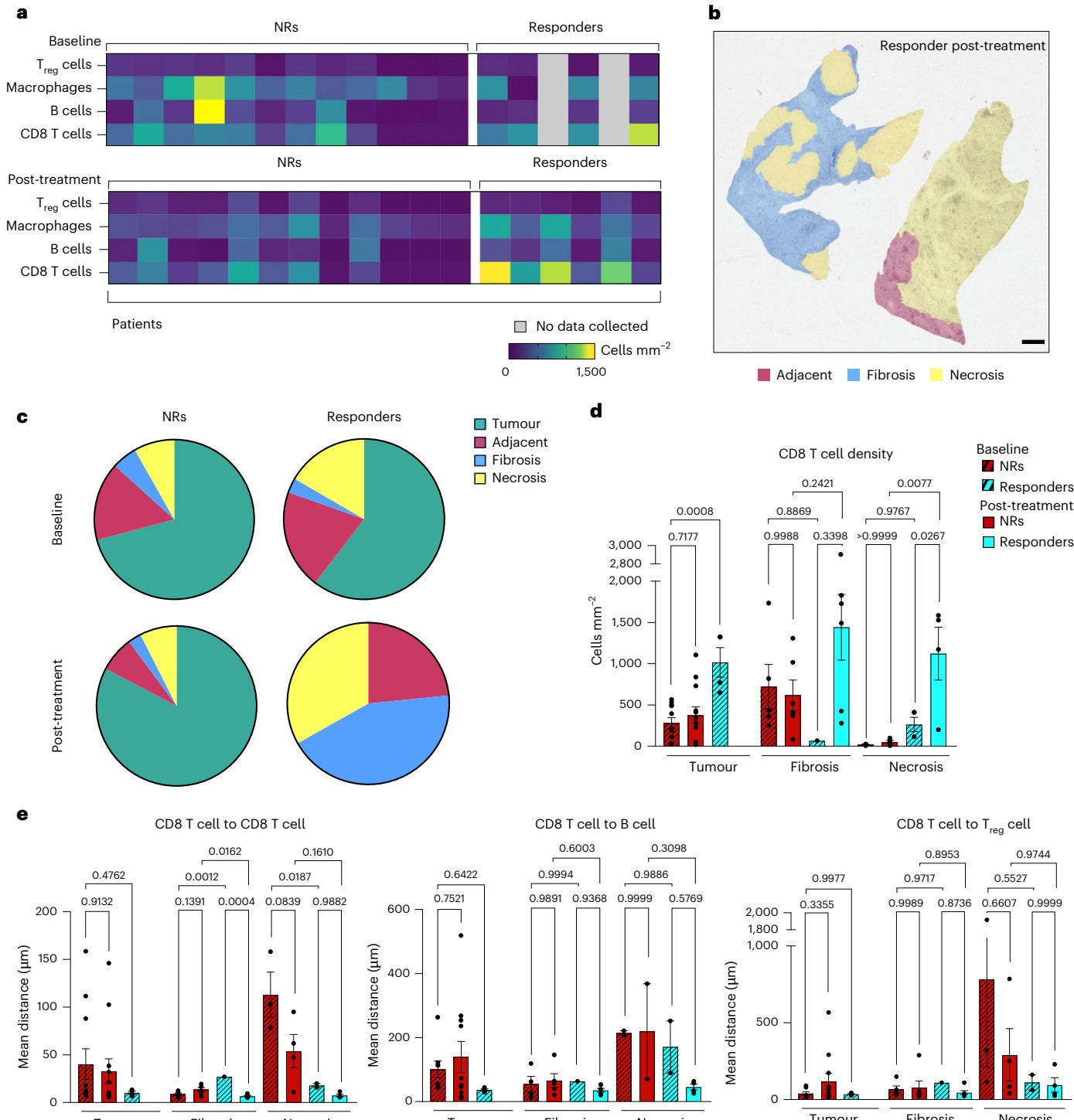

**Fig. 6 | MARQO identifies CD8 T cell enrichment in responders to anti-PD1 treatment. a**, After classification and review, we identified $T_{reg}$ cells ($CD3^+FOXP3^+CD8^-$), macrophages ($CD68^+$), B cells ($CD20^+$) and CD8 T cells ($CD3^+CD8^+$) and evaluated their densities across whole-slide tissues for responders and non-responders (NRs) to cemiplimab treatment at baseline and post-treatment. **b**, Representative image showing the overlay of the tissue with the annotation made by the pathologist using QuPath to demarcate tissue region boundaries, including tumour, adjacent, fibrosis and necrosis, for all tissues used. Scale bar, 500 μm. **c**, Pie charts depicting the tissue compartments annotated by a pathologist for all samples at baseline and post-treatment for responders ($n = 4$ and 6, respectively) and non-responders (NRs) ($n = 12$ for both).

**d**, CD8 T cell densities plotted at baseline and post-treatment for NR and responder groups within tumour ($n = 27$), fibrosis ($n = 18$) and necrosis ($n = 15$) compartments. Data are presented as mean ± s.e.m. Multigroup analyses of variance were performed using one-way analysis of variance followed by Šídák one-way comparison tests. **e**, Bar plots depicting the means of the shortest distances from CD8 T cells to themselves (left), to B cells (middle) and to $T_{reg}$ cells (right) at baseline and post-treatment for non-responders and responders within tumour ($n = 27$), fibrosis ($n = 18$) and necrosis ($n = 15$) compartments. Data are presented as mean ± s.e.m. Multigroup analyses of variance were performed using one-way analysis of variance followed by Šídák one-way comparison tests.

staining as binary outputs, and cell identification and segmentation based on nuclear detection with artificial cytoplasmic extension. Intensity scoring could be considered for appropriate staining protocols in future versions, based on expected tier differences during selection of similarly stained cell clusters. In addition, extracellular matrix evaluation and alternate segmentation protocols could be integrated based on membrane markers when available. We are also trying to reduce the number of falsely determined double-positive cells (doublets) outputted from our classification technique. By calculating frequencies of respective singlet positive cells, singlet negative cells and proximity intensity values for the markers of interest, we can determine a probability assignment and redefine the doublet as a single-marker-stained cell; we also plan to do this for cells with more than two markers co-expressed. These areas of improvement, and additional tools to further analyse multiplex imaging data, are our current targets for future deployments and updates to the available pipeline, which can easily be appended as additional modules in our application.

We envision that MARQO will drive quantitative discoveries in imaging assays. We used MARQO to quantify enrichment and localization of CD8 T cells in HCC responders to neoadjuvant cemiplimab. These findings elucidate previous hypotheses about the interactions of CD8 T cells and will help us better understand mechanisms of response or resistance to immunotherapy[34]. Spatial biology is emerging as a pivotal field for biomarker discovery. By understanding tissue organization, immune cell infiltration and antigen presentation heterogeneity in a harmonized manner, researchers can interpret data across multiple studies and platforms, ultimately developing actionable predictive tests[36–39]. With MARQO, we hope pathologists and researchers will have a necessary tool to quantitatively analyse whole-slide imaging data, augmenting future discoveries and bolstering the field of cancer immunology.

## Methods

### MARQO modules

**Operating the MARQO pipeline.** The MARQO pipeline can be deployed via container technologies, such as Docker and Singularity. It builds on Tensorflow's prebuilt docker image, with a Python virtual environment containing all necessary libraries. The container runs on virtually any set-up: single or multiple cores, computers, servers, high-performance computing clusters or personal machines. Users can interact through a web-based GUI or command line interface, using a semi-automatic approach to fine-tune parameters for both launching and reviewing features (Supplementary Videos 1 and 2).

In the launching tab, users select the imaging modality and initiate tissue masking, preliminary registration, tiling (image decomposition) and colour deconvolution. Although automated, these steps are user-modifiable for optimal quality control. For tissue masking, users can manually draw regions to override automation. During tiling, users define overlap and edge exclusion settings, and the minimum tissue content per tile. By default, tissue is split into 1,000 pixel × 1,000 pixel tiles (~500 μm × 500 μm)—a balance between analysis count and computational load (Supplementary Fig. 2c). The HistomicsTK channel deconvolution module[40], which implements a revised method from ref. 41, dynamically extracts three colour channels using a deconvolution matrix derived from the thumbnail image. For MICSSS, this includes the chromogen stain, nuclear haematoxylin counterstain and a residual channel. Outputs from each step are saved in a configuration file to enable reproducibility in future runs.

In the reviewing tab, users can classify cell clusters, stitch registered whole-slide images into RGB or multichannel formats and reconcile compartment annotations to analyse cell infiltration in designated regions. These annotations come from a standardized '.geojson' file, currently created manually with tools such as QuPath. Future versions aim to integrate self-supervising annotation tools (for example, UNI[42]). Reconciliation populates the final summary table with the tissue

compartment for each cell. Users can visualize cell subsets by toggling marker combinations in the GUI and obtain quantification and density metrics for full tissue or specific compartments. Finally, MARQO outputs can be converted to third-party-compatible formats for further analysis and figure-ready overlay production (Supplementary Fig. 8a and Supplementary Table 1).

**MARQO whole-slide registration module.** MARQO performs a series of registration steps for technologies requiring alignment to enable multiplex cell-resolution quantification. During initial quality control to launch a job, users perform a preliminary translational registration of low-resolution thumbnail images, yielding relative tissue alignment across staining iterations. After tiling, batches of tiles have 10% overlapping borders with neighbouring batches to permit registration and segmentation near tile edges without cropping cells. This overlap was optimized to enable seamless stitching while minimizing redundant computations. Per batch, the MARQO registration module deconvolutes tiles to extract nuclear counterstains, which remain consistent across stains, unlike tissue morphology or positive signals. The deconvolution matrix is dynamically determined during the launch step. MARQO then applies affine and elastic registrations to align all $n$th-stained tiles to the first-stained tile using the SimpleElastix[43] registration package parameters optimized on MICSSS images. The resulting vector field is applied to each tile's red, green and blue channels, enabling cell-resolution registration for each tile batch.

**MARQO segmentation module.** For each registered tile batch, MARQO enacts its segmentation module to identify cells, enabling subsequent metadata extraction in the quantification module. For technologies with iterative nuclear stains such as MICSSS, it performs semantic nuclear segmentation using StarDist—a pretrained algorithm that takes RGB IHC images as input and outputs nuclear masks based on haematoxylin expression[26]. StarDist requires training model metadata, which is included in MARQO's Docker environment. To reconcile multiple segmentation masks per tile batch, MARQO iterates across each nuclear object. If the centroid remains consistent within a tunable threshold (default is 60% of stains), the object is retained in the final composite mask. Otherwise, it is considered an artefact or a lost cell due to tissue damage during staining. For technologies with only one nuclear stain, MARQO performs a single segmentation and uses this nuclear mask for downstream quantification. StarDist is used for singleplex IHC with haematoxylin staining, while a separate StarDist model pretrained on the DAPI channel is used for mIF staining, also yielding semantic nuclear masks. Whether analysing multiplex or singleplex data, MARQO ensures each cell is segmented only once in the final mask. As an alternative, users may opt for CellPose[44], another pretrained deep-learning-based tool for nuclear segmentation[43].

**MARQO quantification module.** In its quantification module, MARQO imports the nuclear semantic mask and expands nuclear boundaries by a user-defined number of pixels per stain to simulate an artificial cytoplasm. The module then extracts pixel-level and morphological metadata for each cell's nucleus, cytoplasm and membrane. These include signal intensity values for the chromogen and counterstain (minimum, maximum, percentiles, standard deviation and more), and perimeter, area, and major and minor axes. The median nuclear-staining intensity is also quantified to help infer cell identity; for instance, RBCs lack nuclei but retain nuclear counterstain. To avoid duplication from overlapping tiles, metadata are only collected from nuclei centroids located within the bounds of the original, unextended tile. Metadata from all tile batches are then appended to a features table containing information for all registered and segmented cells across all stains in the sample.

MARQO also generates a .geojson file containing spatial metadata for all segmented nuclei and cytoplasms. Users may choose to use MARQO exclusively for registration and segmentation, and then import

the raw images and spatial files into external tools such as QuPath to classify cells using alternative approaches, such as supervised neural networks.

**MARQO classification modules.** MARQO provides two classification options, both using unsupervised learning to cluster all segmented and identified cells from the sample into a predefined number of unique cell clusters per marker. After either method, users determine which clusters contain true-positive or true-negative cells using the review GUI. To support this decision, the application overlays cell centroids per cluster onto the raw images and allows visualization of any tile the user chooses. Users may also manually select individual cells as positive or negative. Once all markers are evaluated, the summary table is updated to reflect marker status for each cell. This process can be performed per sample or across multiple samples simultaneously.

The first classification option applies principal component analysis to reduce the dimensionality of the features table. Components explaining 90% of the variance are used to cluster cells into the desired number of clusters via the mini-batch $k$-means algorithm. This faster method supports a reclassify feature within the GUI, allowing users to further subdivide clusters they judge to be heterogeneous. Due to its speed and dimensionality reduction, this approach is recommended for large datasets or those requiring rapid reclustering.

The second option randomly selects 50,000 cells from the features table, applies principal component analysis, and then uses uniform manifold approximation and projection. The resulting low-dimensional projection is clustered using a Gaussian mixed model (GMM), seeded by initial centroids. Approximately 60 GMMs are generated per marker across batches and then merged into the final cluster count by applying $k$-means clustering to the GMM means of each standardized feature. A total of 60 GMMs were empirically determined to best capture the variance in our datasets.

### Statistics and reproducibility

For each figure panel presented, the number of human tissue samples used is indicated in the corresponding figure caption. All immunostaining markers were previously validated and optimized for each specific tissue type. Following optimization, staining was performed once per sample. Reproducibility was ensured through the inclusion of multiple independent samples per experimental condition as biological replicates to support the robustness of the findings.

### Human participants

HCC liver samples used for MICSSS and singleplex IHC were obtained via a single-arm, open-label, phase 2 trial of patients with HCC with resectable tumours (ClinicalTrials.gov, NCT03916627, cohort B). A total of 20 patients were enrolled and received 2 cycles of cemiplimab before surgical resection, as described in the clinical trial publication, including the full protocol provided in the supplementary materials[25]. Core needle biopsies and tumour tissues were obtained from these patients undergoing surgical resection at Mount Sinai Hospital, after obtaining informed consent in accordance with a protocol reviewed and approved by the Institutional Review Board at the Icahn School of Medicine at Mount Sinai (IRB 18-00407). Tonsil samples used for singleplex IHC were obtained from patients undergoing tonsillectomies, from Leica Biosystems, which purchased FFPE blocks from the Deer Park local hospital. NSCLC resection samples used for MICSSS and mIF were obtained from treatment-naive patients undergoing surgical resection at Mount Sinai Hospital, after obtaining consent in accordance with a protocol reviewed and approved by the Institutional Review Board at the Icahn School of Medicine at Mount Sinai (IRB 21-01308). The NSCLC, head and neck squamous cell carcinoma, colorectal cancer, breast cancer, epithelial ovarian cancer, pancreatic ductal adenocarcinoma, glioblastoma, renal cell carcinoma and melanoma samples used for the TMA MICSSS analysis were obtained

by the Cooperative Human Tissue Network in a de-identified fashion without the possibility of being linked with metadata. All human data were either from properly consented patients or anonymized before uploading to our cluster database.

### Sample preparation for singleplex IHC, MICSSS and TMA

For MICSSS and singleplex IHC, pretreatment HCC core biopsies and post-treatment surgically resected HCC lesions were obtained from FFPE blocks. Tissues used to create the TMA were similarly processed, and 1.0-mm diameter punches were made. FFPE tissue sections were sliced at 4 μm. Whole-slide tissue sections were stained either by singleplex IHC or by using the MICSSS protocol as previously described[5]. The TMA was stained with MICSSS. All slides went through an automated immunostainer (Leica Bond RX; Leica Biosystems) that performed baking, chromogenic revelation and nuclear counterstaining. For the singleplex IHC, tonsil samples were subjected to chromogenic revelation via 3,3′-DAB (Vector Laboratories), and tumour samples and all MICSSS samples underwent chromogenic revelation via AEC (Vector Laboratories). All IHC assays were counterstained with haematoxylin. Then, all slides were mounted with a glycerol-based mounting medium and scanned to obtain digital images using an Aperio AT2 scanner and Aperio ImageScope DX visualizer software v.12.3.3 (Leica). For MICSSS, after each round of staining and scanning, slide coverslips were removed in hot water (~50 °C) and tissue sections were bleached. This process was repeated for the length of the panel. Primary antibodies are presented in Supplementary Table 2 for the singleplex IHC, TMA, whole-slide resections and biopsies.

### Multiplex immunofluorescence staining and imaging on COMET

COMET mIF, or automated hyperplex immunofluorescence, staining and imaging was performed on the COMET platform (Lunaphore Technologies). Slides from FFPE blocks were cut at 4 μm and underwent 10 cycles of iterative staining and imaging, followed by an elution of the primary and secondary antibodies[9,45]. In brief, slides were pre-processed with PT Module (Epredia) with Dewax and HIER Buffer H (TA999-DHBH; Epredia) for 60 min at 102 °C. Subsequently, slides were rinsed and stored in Multistaining Buffer (BU06; Lunaphore Technologies) until use. The 20-plex protocol template was generated using the COMET Control Software, and reagents were loaded onto the device to perform the sequential immunofluorescence (seqIF) protocol. A list of primary antibodies with corresponding dilution and incubation times is enclosed in Supplementary Table 3. Secondary antibodies were used as a mix of two species-complementary antibodies. The nuclear signal was detected with DAPI (1:1,000 dilution; catalogue number 62248; Thermo Scientific) after 2 min of dynamic incubation. All reagents were diluted in Multistaining Buffer (BU06; Lunaphore Technologies). For each cycle, the following exposure times were used: 50 ms for DAPI, 400 ms for tetramethylrhodamine isothiocyanate (TRITC) and 200 ms for Cy5. The elution step lasted 2 min for each cycle and was performed with elution buffer (BU07-L; Lunaphore Technologies) at 37 °C. The quenching step lasted for 30 s and was performed with quenching buffer (BU08-L; Lunaphore Technologies). The imaging step was performed with imaging buffer (BU09; Lunaphore Technologies). The seqIF protocol in COMET resulted in a multilayer '.ome.tiff' file, where the imaging outputs from each cycle were stitched and aligned. COMET ome.tiff contains a DAPI image, intrinsic tissue autofluorescence in the TRITC and Cy5 channels, a single fluorescence layer per marker and a single layer per additional image post-elution. Antibody titration was optimized to identify the best antibody dilution and incubation time. Imaging was performed on unstained tissue after each cycle of biomarker staining and antibody elution. The images were used to assess the staining quality and the elution efficiency for each staining condition.

## Image analysis performed on QuPath

To evaluate MARQO's performance, we analysed tiles from resected tissue images generated by singleplex IHC, COMET mIF and MICSSS TMA images using QuPath, a third-party analysis tool commonly used in pathology laboratories[19]. For singleplex IHC and each MICSSS image, colour deconvolution was performed to separate stain vectors—haematoxylin, AEC or DAB chromogens and a residual channel. Stain vectors were estimated from a selected ROI containing balanced positive and negative cells, along with a blank area to avoid downsizing artefacts that could affect deconvolution quality. This step was not needed for COMET mIF .ome.tiff images, where each stain was already assigned to a separate channel. Whole-tissue annotations were created using QuPath's 'simple tissue detection' to distinguish tissue from background glass. For TMA images, separate annotations were made for each core. Individual cells were identified using StarDist, a pretrained nuclear segmentation algorithm that outputs nuclear masks based on haematoxylin or DAPI expression and expands these to include cytoplasmic and membranous compartments using a user-defined diameter. Intensity values for haematoxylin and chromogens, along with morphological features, were recorded for nuclear, cytoplasmic and total cell compartments.

For each biomarker and dataset, a machine learning classifier was trained using intensity and morphological values from manually selected positive and negative cells. This classifier was applied to all images stained for the same biomarker across the cohort to identify positive cells. Due to the heterogeneous nature of the tissue types in the TMA, high-level classifiers applied across all cores were sometimes insufficient. In such cases, tissue-specific classifiers trained and applied within individual tissue types improved identification accuracy. Finally, cell marker density data, calculated as the number of positive cells per total tissue area, were exported[7].

## Manual cell counting for validation

We compared MARQO's nuclear segmentation performance to that of a trained pathologist manually selecting individual cells. A pathologist selected 15 post-registered tiles (1,000 pixels × 1,000 pixels, ~500 μm × 500 μm) from FOXP3-stained images in the human HCC cohort. FOXP3 stains were chosen due to minimal AEC interference with the haematoxylin counterstain. Tiles represented diverse tissue architectures and cell densities. Using QuPath, the pathologist created a 'Cell' class and used the 'Points' tool to click within each nucleus, generating cell count and coordinate data, which were exported as individual .geojson files. We then compared these with MARQO's output using a point-in-polygon strategy. Manually selected points inside MARQO-segmented nuclear boundaries were labelled 'MARQO+, manual+', with matched cells removed from further analysis. Remaining points outside any segmentation were labelled 'MARQO−, manual+', and unmatched MARQO cells were labelled 'MARQO+, manual−'.

Of the 15 tiles, 5 included large numbers of RBCs to test MARQO's RBC filtering. The pathologist marked all perceived RBCs using the Points tool in QuPath. Points inside MARQO's nuclear boundaries were classified as 'miss', while those outside were 'hit', indicating successful filtering.

For classification validation, 34 post-registered tiles (1,000 pixels × 1,000 pixels) were selected across various stains and tissue types, including tumour-adjacent and HCC tissues, and markers CD3, FOXP3, CD68 and PanCK. These tiles also varied in tissue architecture and cell density. The pathologist created a 'positive' class and clicked points within nuclei of cells deemed positive, exporting the metadata as .geojson files. We again applied the point-in-polygon strategy. Points inside MARQO-identified positive nuclei were labelled 'MARQO+, manual+'. Positive MARQO cells without matching points were 'MARQO+, manual−', and negative MARQO cells without any clicked points were 'MARQO−, manual−'. Negative MARQO cells with a clicked point were labelled 'MARQO−, manual+'. Manually selected points outside any segmented cell boundary were excluded from analysis.

## Imaging proximity analysis

Using the features table, we extracted the centroid coordinates of all cells and stratified them by marker combination patterns and localization within annotated tissues. For each cell in population A, we computed the distance to the nearest neighbour in population B. From the resulting distribution of minimal distances, we computed a Gaussian kernel density estimation curve and estimated the mode. This process was repeated for all possible pairs between cell types A and B for each patient sample, stratified per region of interest. Finally, we plotted the distribution of modes from the CD8 T cell perspective to themselves, B cells and $T_{reg}$ cells across tissue regions. This analysis will be available as a review module in a future deployment of the pipeline.

## Reporting summary

Further information on research design is available in the Nature Portfolio Reporting Summary linked to this article.

## Data availability

Image files are available upon request from the corresponding author. Patient co-expression and cell type data are provided in Supplementary Table 4.

## Code availability

A container with MARQO's GUI to analyse samples locally is available at https://github.com/igorafsouza/MARQO. All source code is available from the corresponding author upon reasonable request[46]. Requests for service to run larger samples on a cluster should be made by contacting the corresponding author.

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

## Acknowledgements

We thank the patients and their families for participating in our research study and for providing clinical trial and donor specimens. We thank the Biorepository and Pathology CoRE Laboratory of the Icahn School of Medicine at Mount Sinai for support. We also thank members of the Merad and Gnjatic laboratories for their help and support. This work was supported in part through the computational and data resources and staff expertise provided by Scientific Computing and Data at the Icahn School of Medicine at Mount Sinai and supported by the Clinical and Translational Science Awards grant UL1TR004419 from the National Center for Advancing Translational Sciences. The clinical trial (ClinicalTrials.gov, NCT03916627, cohort B) and part of this project were funded by Regeneron. S.G. was partially supported by National Institutes of Health (NIH) grants CA224319, DK124165, CA263705 and CA196521. M.M. was partially supported by NIH grants CA257195, CA254104 and CA154947. T.U.M. was partially supported by the Tisch Cancer Institute Cancer Center Support Grant (P30 CA196521).

## Author contributions

M.B., P.H., A.T., S.H., M.M. and S.G. conceptualized the project. T.U.M., M.M. and S.G. obtained funding for the project. M.B., G.I., P.H. and S.G. designed the experiments. M.B., I.F., G.I., V.R., E.G.-K., S.O., R.C., L.T., J.L.B., S.H. and G.A. performed the experiments. M.B., I.F., G.I., P.H. and S.G. analysed the experiments. M.S. and T.U.M. provided all clinical care to patients in the clinical trial. S.O., R.C., Z.Z., S.C.W, M.I.F. and R.B. provided expertise for pathological response assessment and tissue annotation. C.H. coordinated the clinical and research teams. M.B. and I.F. developed the pipeline. M.B., I.F. and V.R. performed computational analysis. A.T., E.G.-K., S.K.-S., M.M. and S.G. provided intellectual input. M.B. wrote the paper. P.H. and S.G. edited the paper. All authors provided feedback on the draft of the paper.

## Competing interests

M.M. serves on the scientific advisory board and holds stock from Compugen, Dynavax, Morphic Therapeutic, Asher Bio, Dren Bio, Nirogy, Oncoresponse and Owkin. M.M. serves on the scientific advisory board of Innate Pharma, DBV and Genenta. M.M. receives funding for contracted research from Regeneron and Boehringer Ingelheim. T.U.M. has served on advisory and/or data safety monitoring boards for Rockefeller University, Regeneron Pharmaceuticals, Abbvie, Bristol Meyers Squibb, Boehringer Ingelheim, Atara, AstraZeneca, Genentech, Celldex, Chimeric, Glenmark, Simcere, Surface, G1 Therapeutics, NGMbio, DBV Technologies, Arcus and Astellas, and has research grants from Regeneron, Bristol Myers Squibb, Merck and Boehringer Ingelheim. S.G. reports past consultancy or advisory roles for Merck and OncoMed; research funding from Regeneron Pharmaceuticals related to this study, and research funding from Boehringer Ingelheim, Bristol Myers Squibb, Celgene, Genentech, EMD Serono, Pfizer and Takeda, unrelated to the current work. S.G. is a named co-inventor on an issued patent (US20190120845A1) for multiplex immunohistochemistry to characterize tumours and treatment responses. The technology is filed through Icahn School of Medicine at Mount Sinai and is currently unlicensed. This technology was used to evaluate tissue in this study and the results could impact the value of this technology.

## Additional information

**Correspondence and requests for materials** should be addressed to Sacha Gnjatic.

**Mark Buckup** [1,2,3], **Igor Figueiredo**[1,2,3,8], **Giorgio Ioannou** [1,2,3,8], **Sinem Ozbey**[1,2,3], **Rafael Cabal**[1,2,3], **Alexandra Tabachnikova**[1,2,3], **Leanna Troncoso** [1,2,3], **Jessica Le Berichel**[1,2,3], **Zhen Zhao**[4], **Stephen C. Ward**[4], **Clotilde Hennequin**[1,2,3], **Guray Akturk** [1,2,3], **Steve Hamel**[1,2,3], **Maria Isabel Fiel**[4], **Rachel Brody**[4], **Myron Schwartz** [5], **Thomas U. Marron** [1,2,3,6], **Seunghee Kim-Schulze** [1,2,3,7], **Vladimir Roudko**[1,2,3], **Edgar Gonzalez-Kozlova**[1,2,3], **Pauline Hamon** [1,2,3], **Miriam Merad** [1,2,3,7] & **Sacha Gnjatic** [1,2,3,4,6,7] ✉

[1]Marc and Jennifer Lipschultz Precision Immunology Institute, Icahn School of Medicine at Mount Sinai, New York, NY, USA. [2]The Tisch Cancer Institute, Icahn School of Medicine at Mount Sinai, New York, NY, USA. [3]Department of Immunology and Immunotherapy, Icahn School of Medicine at Mount Sinai, New York, NY, USA. [4]Department of Pathology, Molecular and Cell-Based Medicine, Icahn School of Medicine at Mount Sinai, New York, NY, USA. [5]Department of Surgery, Icahn School of Medicine at Mount Sinai, New York, NY, USA. [6]Division of Hematology/Oncology, Icahn School of Medicine at Mount Sinai, New York, NY, USA. [7]Human Immune Monitoring Center, Icahn School of Medicine at Mount Sinai, New York, NY, USA. [8]These authors contributed equally: Igor Figueiredo, Giorgio Ioannou. ✉e-mail: sacha.gnjatic@mssm.edu

# Reporting Summary

## Statistics

For all statistical analyses, confirm that the following items are present in the figure legend, table legend, main text, or Methods section.

| n/a | Confirmed | |
|---|---|---|
| ☐ | ☒ | The exact sample size (*n*) for each experimental group/condition, given as a discrete number and unit of measurement |
| ☐ | ☒ | A statement on whether measurements were taken from distinct samples or whether the same sample was measured repeatedly |
| ☐ | ☒ | The statistical test(s) used AND whether they are one- or two-sided *Only common tests should be described solely by name; describe more complex techniques in the Methods section.* |
| ☐ | ☒ | A description of all covariates tested |
| ☐ | ☒ | A description of any assumptions or corrections, such as tests of normality and adjustment for multiple comparisons |
| ☐ | ☒ | A full description of the statistical parameters including central tendency (e.g. means) or other basic estimates (e.g. regression coefficient) AND variation (e.g. standard deviation) or associated estimates of uncertainty (e.g. confidence intervals) |
| ☐ | ☒ | For null hypothesis testing, the test statistic (e.g. *F*, *t*, *r*) with confidence intervals, effect sizes, degrees of freedom and *P* value noted *Give P values as exact values whenever suitable.* |
| ☒ | ☐ | For Bayesian analysis, information on the choice of priors and Markov chain Monte Carlo settings |
| ☐ | ☒ | For hierarchical and complex designs, identification of the appropriate level for tests and full reporting of outcomes |
| ☐ | ☒ | Estimates of effect sizes (e.g. Cohen's *d*, Pearson's *r*), indicating how they were calculated |

*Our web collection on statistics for biologists contains articles on many of the points above.*

## Software and code

Policy information about availability of computer code

| Data collection | Aperio ImageScope DX visualizer software v.12.3.3 (Leica) was used for MICSSS and singleplex IHC image acquisition, COMET™ Viewer 1.0.2 (Lunaphore) used to visualize COMET OME-TIFF images |
|---|---|
| Data analysis | QuPath (Open source software) v0.5.1 was used for comparative image analysis. A container with MARQO's GUI to analyze samples locally is available at https://github.com/igorafsouza/MARQO. All source code is available from the corresponding author upon reasonable request. Requests for service to run larger samples on a cluster should be made by contacting the corresponding author. A list of package dependencies present in our environment are listed in our GitHub (marqo-v1.0.0/ requirements.txt). |

For manuscripts utilizing custom algorithms or software that are central to the research but not yet described in published literature, software must be made available to editors and reviewers. We strongly encourage code deposition in a community repository (e.g. GitHub). See the Nature Portfolio guidelines for submitting code & software for further information.

# Data

Policy information about availability of data

All manuscripts must include a data availability statement. This statement should provide the following information, where applicable:

- Accession codes, unique identifiers, or web links for publicly available datasets
- A description of any restrictions on data availability
- For clinical datasets or third party data, please ensure that the statement adheres to our policy

> We have provided deidentified sample images per imaging technology on our GitHub repository. Because specimens related to trials require study sponsor permission, the remaining image files we analyzed are available upon request from the corresponding author. Patient co-expression and cell type data are provided in Extended Data Table 1.

# Research involving human participants, their data, or biological material

Policy information about studies with human participants or human data. See also policy information about sex, gender (identity/presentation), and sexual orientation and race, ethnicity and racism.

**Reporting on sex and gender**

> Patients were recruited into the clinical trial (ClinicalTrials.gov, NCT03916627, Cohort B) regardless of sex or gender. All patients whom are deemed to be candidates for surgery and fit inclusion/exclusion criteria are offered participation in the clinical trial.
> All patients at Mount Sinai Hospital undergoing surgery regardless of any prior treatment received are asked to participate in the biorepository, a standard practice at all large academic hospitals.

**Reporting on race, ethnicity, or other socially relevant groupings**

> This study did not use race, ethnicity, or other socially relevant grouping. Patients grouped by their tumor location were obtained in a deidentified fashion without the possibility of being linked with metadata. Patients enrolled in the clinical trial were grouped by treatment response.

**Population characteristics**

> All samples from patients outside the clinical trial were obtained deidentified without the possibility of being linked with metadata. This single-centre, open-label, single-arm, phase 2 trial of cemiplimab monotherapy administered before and after definitive surgery enrolled patients with early-stage hepatocellular carcinoma. Eligible patients were aged 18 years or older and had confirmed resectable hepatocellular carcinoma (Liver Imaging Reporting and Data System [LIRADS] score of 5 on imaging or biopsyproven tumour, or both), an Eastern Cooperative Oncology Group performance status of 0 or 1, and adequate liver function. Patients were enrolled regardless of the underlying cause of hepatocellular carcinoma; patients with a history of hepatitis C virus (HCV) or hepatitis B virus (HBV) infection were eligible for enrolment if viral clearance had occurred or circulating virus was suppressed on HBV-directed therapies. Patients with HIV with an undetectable viral load by PCR and a CD4+ T-cell count higher than 350 cells per µL were also eligible for enrolment.

**Recruitment**

> Criteria of recruitment into the clinical trial (NCT03916627) are listed below, and also found in the protocol which is a supplement to the previously published clinical study (Marron et al, Lancet GI/Hep 2022). Patients enrolled in the biorepository do not have to meet any prespecified inclusion/exclusion criteria, however, retrospectively assessing their charts they all do, also, meet these criteria:
>
> Key Inclusion Criteria:
> • Patient must have a known diagnosis of HCC as defined in the protocol
> • Patient must be willing and able to provide blood samples at the indicated time points
> • Patient must be willing and able to have excisional or core needle biopsies of tumor prior to initiation of cemiplimab as defined in the protocol
> • Eastern Cooperative Oncology Group (ECOG) performance status of 0 or 1
> • Patient is determined to be a surgical candidate for resection of their tumor
> • Adequate organ and bone marrow function as defined in the protocol
>
> Key Exclusion Criteria:
> • Patients who have had any systemic anti-cancer therapy or radiotherapy within 6 months prior to entering the study for their current tumor or a different primary tumor
> • Patients whose tumor burden, or pace of tumor growth, in the opinion of the Investigator will not permit delaying surgery
> • Patients who have participated in a study of an investigational agent or an investigational device within 4 weeks of study therapy or 5 half-lives (whichever is longer)
> • Patients who have had major surgery within 14 days prior to initiation of neoadjuvant therapy
> • Patients with metastatic disease for whom the intent of surgery would not be curative
> • Uncontrolled, intercurrent illness as defined in the protocol and as determined by the Investigator
> • Is receiving systemic steroid therapy or any other form of immunosuppressive therapy within 7 days prior to the first dose of study treatment
> • Has active autoimmune disease that has required systemic treatment in the past 1 year
> • Has a known, additional malignancy that is progressing and/or requires active treatment. Exceptions include patients with: basal cell carcinoma of the skin or squamous cell carcinoma of the skin that has undergone potentially curative therapy; in situ cervical or anal cancer; prostate cancer on stable dose of hormonal therapy without rising PSA; breast cancer who have been treated with curative intent, who may be on hormonal therapy.
> • Encephalitis, meningitis, or uncontrolled seizures in the year prior to informed consent
> • Uncontrolled infection with human immunodeficiency virus (HIV), HBV or hepatitis C infection (HCV); or diagnosis of immunodeficiency as defined in the protocol

Tonsil samples were obtained from patients undergoing tonsillectomies, from Leica Biosystems, which purchased FFPE blocks from the Deer Park local hospital. Non-small cell lung cancer (NSCLC) resection samples were obtained from treatment-naive patients undergoing surgical resection at Mount Sinai Hospital (New York, NY). NSCLC, head and neck squamous cell carcinoma (HNSCC), colorectal cancer (CRC), breast cancer (BC), epithelial ovarian cancer (EOC), pancreatic ductal adenocarcinoma (PDAC), glioblastoma (GBM), renal cell carcinoma (RCC), and melanoma (MEL) samples were obtained by the Cooperative Human Tissue Network.

Ethics oversight | Institutional Review Board at the Icahn School of Medicine at Mount Sinai (IRB Human Subjects Electronic Research Applications 18-00407 and 21-01308)

Note that full information on the approval of the study protocol must also be provided in the manuscript.

# Field-specific reporting

Please select the one below that is the best fit for your research. If you are not sure, read the appropriate sections before making your selection.

☒ Life sciences          ☐ Behavioural & social sciences          ☐ Ecological, evolutionary & environmental sciences

For a reference copy of the document with all sections, see nature.com/documents/nr-reporting-summary-flat.pdf

# Life sciences study design

All studies must disclose on these points even when the disclosure is negative.

Sample size | With 21 patients in cohort B, the cohort will test a null hypothesis of poor overall response of 5% or less versus an alternative hypothesis of a promising response rate of 20% or more, at a 10% one-sided significance level and 80% power. If the number of pathological responders is 3 or more in a cohort, the null hypothesis is rejected and the treatment for that cohort is recommended for further study. The additional patient samples size that have been analyzed in this manuscript was determined as sufficient for the validation of the different MARQO application.

Data exclusions | Patients were excluded from enrolment if they had metastatic disease, if the surgery was not expected to be curative, or if they had a known additional malignancy requiring active treatment. Patients could not be receiving chronic systemic immunosuppression or have active autoimmune disease requiring systemic treatment in the past year, except for patients with endocrinopathies on hormone replacement therapy. Pregnant women and patients who had undergone a transplant were excluded, as were any patients with a history of CNS or pulmonary inflammatory conditions.

Replication | Reproducibility measures are not applicable in this context, as the study represents a discrete and completed clinical trial. All analyses were conducted on a single, defined cohort of patients with HCC treated with cemiplimab. Additional cohorts incorporating immunodynamic interventions to the cemiplimab treatment are currently being enrolled, but these are part of separate ongoing studies and will be reported in future publications.

Randomization | The clinical trial was a single-arm study with no randomization. Other sample obtainment and assignment was random.

Blinding | Blinding was not performed; the clinical trial was a open-label single-arm study.
Regarding the other samples, blinding was not relevant as each sample was assigned based on the imaging technology and direct comparison required for MARQO validation.

# Reporting for specific materials, systems and methods

We require information from authors about some types of materials, experimental systems and methods used in many studies. Here, indicate whether each material, system or method listed is relevant to your study. If you are not sure if a list item applies to your research, read the appropriate section before selecting a response.

## Materials & experimental systems

| n/a | Involved in the study |
|-----|----------------------|
| ☐ ☒ | Antibodies |
| ☒ ☐ | Eukaryotic cell lines |
| ☒ ☐ | Palaeontology and archaeology |
| ☒ ☐ | Animals and other organisms |
| ☐ ☒ | Clinical data |
| ☒ ☐ | Dual use research of concern |
| ☒ ☐ | Plants |

## Methods

| n/a | Involved in the study |
|-----|----------------------|
| ☒ ☐ | ChIP-seq |
| ☒ ☐ | Flow cytometry |
| ☒ ☐ | MRI-based neuroimaging |

# Antibodies

| | |
|---|---|
| Antibodies used | All antibodies are commercially available and have been validated by the vendor for the use of immunohistochemistry or immunofluorescence. Data are available on the manufacturer's website. |

| | |
|---|---|
| Validation | Validation Statements MICSSS: |

CD3 LN10 Leica Biosystems: Immunohistochemical analysis of paraffin-embedded human skin with mycosis fungoides (website)
PDL1: Immunohistochemical analysis of paraffin-embedded human non small cell lung carcinoma (website)
PD1: Immunohistochemical analysis of paraffin-embedded human tonsil (website)
CD8: Immunohistochemical analysis of paraffin-embedded human tonsil (website)
CD3 2GV6 Ventana: Immunohistochemical analysis of paraffin-embedded human tonsil (website)
PanCK: Immunohistochemical analysis of paraffin-embedded human tonsil (website)
FoxP3: Immunohistochemical analysis of paraffin-embedded human tonsil (website)
Ki-67: Immunohistochemical analysis of paraffin-embedded human tonsil (website)
aSMA: Immunohistochemical analysis of paraffin-embedded human liver (website)
CD68: Immunohistochemical analysis of paraffin-embedded human tonsil (website)
CD20: Immunohistochemical analysis of paraffin-embedded human tonsil (website)
MZB1: Immunohistochemical analysis of paraffin-embedded human lymph node (website)

Validation Statements Lunaphore:
CD11c: Immunohistochemical analysis of paraffin-embedded human tonsil (website)
LAG-3: Immunohistochemical analysis of paraffin-embedded human tonsil (website)
CK: Immunohistochemical analysis of paraffin-embedded human tonsil (website)
FoxP3: Immunohistochemical analysis of paraffin-embedded human tonsil (website)
CD3: Immunohistochemical analysis of paraffin-embedded human tonsil (website)
CD8: Immunohistochemical analysis of paraffin-embedded human tonsil (website)
CD11b : Immunohistochemical analysis of paraffin-embedded human spleen (website)
Ki-67 : Immunohistochemical analysis of paraffin-embedded human tonsil (website)
CD68 : Immunohistochemical analysis of paraffin-embedded human tonsil (website)
CD4 : Immunohistochemical analysis of paraffin-embedded human tonsil (website)
CD20 : Immunohistochemical analysis of paraffin-embedded human tonsil (website)
PD-1 : Immunohistochemical analysis of paraffin-embedded human tonsil (website)
CD38 : Immunohistochemical analysis of paraffin-embedded human bone marrow (website)
CD163 : Immunohistochemical analysis of paraffin-embedded human placenta (website)
CD45RA : SeqIF™ (sequential immunofluorescence) staining on formalin-fixed paraffin-embedded human pancreatic carcinoma (website)
CD56 : Immunohistochemical analysis of paraffin-embedded human pancreas (website)
aSMA : Immunohistochemical analysis of paraffin-embedded human colon (website)
Vimentin : Immunohistochemical analysis of paraffin-embedded human melanoma (website)
HLA-DR : Immunohistochemical analysis of paraffin-embedded human tonsil (website)
PD-L1 : Immunohistochemical analysis of paraffin-embedded human lung adenocarcinoma (website)
Alexa FluorTM Plus 555 goat anti-rabbit : Immunofluorescence analysis of MCF 10A (positive model) and T-47D (negative model) cells stained with Vimentin Polyclonal Antibody (Product # PA5-27231) (website)
Alexa FluorTM Plus 647 goat anti-mouse : Immunofluorescence analysis of SH-SY5Y (positive model) and T-47D (negative model) cells stained with Nestin Monoclonal Antibody (10C2), eBioscience™ (Product # 14- 9843-80) (website)
Alexa FluorTM Plus 555 goat anti-mouse : Immunofluorescence analysis of SH-SY5Y (positive model) and T-47D (negative model) cells stained with Nestin Monoclonal Antibody (10C2), eBioscience™ (Product # 14- 9843-80) (website)
Alexa FluorTM Plus 647 goat anti-rabbit : Immunofluorescent analysis of ZO-1 in A549 cells (website)

# Clinical data

Policy information about clinical studies

All manuscripts should comply with the ICMJE guidelines for publication of clinical research and a completed CONSORT checklist must be included with all submissions.

| | |
|---|---|
| Clinical trial registration | NCT03916627 |

| | |
|---|---|
| Study protocol | https://clinicaltrials.gov/ct2/show/NCT03916627?term=cemiplimab&cond=HCC&cntry=US&state=US%3ANY&city=New +York&draw=2&rank=1 |

| | |
|---|---|
| Data collection | Patients were recruited between June 14, 2018 and Nov 25, 2020 at Mount Sinai Hospital, New York. Samples used for this study were also acquired between June 14, 2018 and December 28, 2020. All patients whom are deemed to be candidates for surgery and fit inclusion/exclusion criteria are offered participation in the clinical trial. All patients treated "off-label" prior to the trial opening were enrolled onto the institutional biorepository IRB-approved informed consent. |

| | |
|---|---|
| Outcomes | Primary Outcome Measures: |

1. Major pathologic response (MPR) at time of surgery for the NSCLC cohorts
 [Time Frame: At time of surgery]
 Cohorts A1, A2, A3
2. Significant tumor necrosis (STN) at time of surgery is the primary endpoint for the HCC cohorts
 [Time Frame: At time of surgery]

Cohort B, B2
3. Major treatment effect (MTE) at time of surgery is the primary endpoint for the HNSCC cohort
   [Time Frame: At time of surgery]
   Cohort C

Secondary Outcome Measures:
 1. Delay to surgery
   [Time Frame: Surgery > 28 days following the end of the cycle of last dose of cemiplimab]
   Defined as surgery > 28 days following the end of the cycle of last dose of cemiplimab in the neoadjuvant period.
 2. Event-free survival (EFS)
   [Time Frame: Up to 60 months following surgery]
   Defined as the time from the first dosing of cemiplimab (SBRT for cohort B2) to the date of disease progression that precluded definitive surgery, or recurrence of tumor after successful surgery, or death from any cause.
 3. Disease-free survival (DFS)
   [Time Frame: Up to 60 months following surgery]
   Defined as the time from date of surgery until recurrence of tumor or death from any cause after successful surgery and recovery.
 4. Overall response rate (ORR)
   [Time Frame: Up to 60 months following surgery]
   Defined as the percent of patients with a complete response (CR) or partial response (PR) documented by the Investigator per RECIST 1.1 as described in the protocol.
 5. Overall survival (OS)
   [Time Frame: Up to 60 months following surgery]
   Defined as the time from the first dosing of cemiplimab (chemotherapy for cohort A3 and SBRT for cohort B2) to date of death for any reason.
 6. OS rate
   [Time Frame: 12 months]
 7. OS rate
   [Time Frame: 18 months]
 8. OS rate
   [Time Frame: 24 months]
 9. OS rate
   [Time Frame: 36 months]
 10. OS rate
   [Time Frame: 48 months]
 11. OS rate
   [Time Frame: 60 months]
 12. Incidence of treatment-emergent adverse events (TEAEs)
   [Time Frame: Up to 60 months following surgery]
   Grade 3 or higher per Common Terminology Criteria for Adverse Events (CTCAE V5.0).
 13. Incidence of irAEs
   [Time Frame: Up to 60 months following surgery]
   Grade 3 or higher per CTCAE V5.0.
 14. Incidence of SAEs
   [Time Frame: Up to 60 months following surgery]
   Grade 3 or higher per CTCAE V5.0.
 15. Incidence of deaths
   [Time Frame: Up to 60 months following surgery]
 16. Incidence of laboratory abnormalities
   [Time Frame: Up to 60 months following surgery]
   Grade 3 or higher per CTCAE V5.0.
 17. Change in tumor-infiltrating CD8 T-cell density
   [Time Frame: Baseline to time of surgery]
   Defined as the change from baseline to the time of surgery.

# Plants

Seed stocks

*Report on the source of all seed stocks or other plant material used. If applicable, state the seed stock centre and catalogue number. If plant specimens were collected from the field, describe the collection location, date and sampling procedures.*

Novel plant genotypes

*Describe the methods by which all novel plant genotypes were produced. This includes those generated by transgenic approaches, gene editing, chemical/radiation-based mutagenesis and hybridization. For transgenic lines, describe the transformation method, the number of independent lines analyzed and the generation upon which experiments were performed. For gene-edited lines, describe the editor used, the endogenous sequence targeted for editing, the targeting guide RNA sequence (if applicable) and how the editor was applied.*

Authentication

*Describe any authentication procedures for each seed stock used or novel genotype generated. Describe any experiments used to assess the effect of a mutation and, where applicable, how potential secondary effects (e.g. second site T-DNA insertions, mosiacism, off-target gene editing) were examined.*

