## [Peer Review File · Nature Biomedical Engineering]

Multiparametric cellular and spatial organization in cancer tissue lesions with a streamlined pipeline

Corresponding Author: Dr Sacha Gnjjatic

This manuscript has been previously reviewed at another journal. This document only contains information relating to versions considered at Nature Biomedical Engineering.

Version 0:

Decision Letter:

Dear Dr Gnjjatic,

Thank you again for submitting to *Nature Biomedical Engineering* your manuscript, "MARQO pipeline resolves multiparametric cellular and spatial organization in cancer tissue lesions". The manuscript has been seen by three experts, whose reports you will find at the end of this message.

You will see that the reviewers appreciate the work. However, they express concerns about the degree of benchmarking provided and the wider utility of the pipeline (such as its extensibility), and provide useful suggestions for improvement. We hope that with substantial further work you can address the criticisms and convince the reviewers of the merits of the study.

When you are ready to resubmit your manuscript, please upload the revised files, a point-by-point rebuttal to the comments from all reviewers, the [reporting summary](https://www.nature.com/authors/policies/ReportingSummary.pdf), and a cover letter that explains the main improvements included in the revision and responds to any points highlighted in this decision.

Please follow the following recommendations:

- * Clearly highlight any amendments to the text and figures to help the reviewers and editors find and understand the changes (yet keep in mind that excessive marking can hinder readability).
- * If you and your co-authors disagree with a criticism, provide the arguments to the reviewer (optionally, indicate the relevant points in the cover letter).
- * If a criticism or suggestion is not addressed, please indicate so in the rebuttal to the reviewer comments and explain the reason(s).
- * Consider including responses to any criticisms raised by more than one reviewer at the beginning of the rebuttal, in a section addressed to all reviewers.
- * The rebuttal should include the reviewer comments in point-by-point format (please note that we provide all reviewers will the reports as they appear at the end of this message).
- * Provide the rebuttal to the reviewer comments and the cover letter as separate files.

We expect that you will be able to resubmit the manuscript within 16 weeks of receiving this message. If this is the case, you will be protected against potential scooping. Otherwise, we will be happy to consider a revised manuscript as long as the significance of the work is not compromised by work published elsewhere or accepted for publication at *Nature Biomedical Engineering*.

We hope that you will find the referee reports helpful when revising the work. Please do not hesitate to contact me should you have any questions.

Best wishes,

Pep

Pep Pàmies

Chief Editor, <http://www.nature.com/nbme>>*Nature Biomedical Engineering*

Reviewer #1 (Report for the authors (Required)):

The manuscript "MARQO pipeline resolves multiparametric cellular and spatial organization in cancer tissue lesions" by Mark Buckup et al. addresses the need to analyse imaging data from tissue proteomic assays. In particular, it aims to provide a single platform to cope with various assay protocols and avoid the cumbersome, lengthy, and error-prone habit of using different software packages sequentially. As a use case, the authors demonstrate CD8 T cell enrichment in hepatocellular carcinoma responders to neoadjuvant cemiplimab in a phase II clinical trial.

The workflow and use case are presented clearly and understandably by following well-accepted practices. In particular, the software's documentation for that workflow is beyond average and demonstrates the authors' wish to provide a tool that the community picks up and uses.

The presented workflow is based on well-accepted and known algorithms and aims to facilitate the use and application of known tools rather than the development of new tools. However, the availability of open-source workflows with low entry barriers for spatial proteomics is currently a primary challenge in facilitating a broader usage of these technologies. Thus, the presented workflow is timely and fits the community's needs.

It would benefit the manuscript if the authors outlined how to disseminate the workflow and ensure that it is used and further developed by the community. Also, a strategy for adapting and implementing new protocols and image analysis routines is missing. However, this is critical to guaranteeing the sustainability and longevity of that workflow.

Please find my detailed comments below:

Major :

I. 355 ff. You claim that it is an advantage that the ML algorithms you provide do not need to be retrained. In practice, it turned out that retraining is necessary in some instances. Thus, the workflow design limits the possibility of adapting to such cases. This might render the workflow less valuable in the future. The ability to retrain ML algorithms (e.g. Stardist) should be made possible as an option.

I. 159 ff. The cited literature does not support the claim that a dice score of 0.85 indicates an automated segmentation better than a manual one. It would help if you were careful not to overinterpret your findings.

L 167 ff. It is well-known and accepted that ML introduces a bias in any image analysis. However, it is questionable that a workflow with parameters set by a user is introducing less bias as claimed for the presented workflow. The manuscript will significantly profit from a fairer and more evidence-based discussion.

I. 286 ff. The "semi-supervised" approach presented in your manuscript is useful and provides insightful results. However, it comes with the expense of needing a well-trained expert to run and evaluate the workflow. This aspect is missing in the discussion section.

Minor :

I. 68 The ability to run the analysis in the cloud is highly appreciated. However, pointing out legal issues with patient samples being analysed in the cloud might be necessary to address, at least in the conclusion/discussion section.

I. 87 Comparing a tailored workflow with a generic tool like QuPath seems a bit unfair unless you specifically point out which steps will not be possible in QuPath.

I. 130 Please provide more details on the denoising algorithm and why it is necessary.

I. 140 The decision process when a nuclear mask is rejected is outlined in detail in the MM section. It might be helpful to mention it in the main text.

I. 197 ff. In this application, the workflow is as accurate as the pathologist's manual annotation. However, there might be cases that serve as counter-examples.

I. 297 ff. It might be worth commenting on which of the parameters needed to be fine-tuned to adapt to the different assays and how much expert knowledge is needed to find the right parameters.

I. 452 Ideally, the data will be deposited in a public repository.

Remarks on code availability:

The code is sufficiently documented and contains instruction on how to install and run the software.

Reviewer #2 (Report for the authors (Required)):

Buckup. M. et.al. introduces MARQO; a computational pipeline to streamline the workflow of multiple modules necessary for extracting higher level insights from the cancer tissue lesions over different multiplex immunostaining-based imaging modalities. Namely each of these modules as registration, segmentation, and quantification, are deeply intertwined in terms of their parameters and outputs and are pivotal parts of multiplexed immunohistochemistry imaging studies. As such, they have been quite heavily studied and matured to varying degrees. The bold offering as the integration of all these modules topped with an even more preferential yet necessary analytical one as the main thesis of this study, is a meaningful step toward more efficient knowledge elucidation of the underlying images. Also providing the possibility for the users to interact and influence the output of these modules through a graphical user interface is a great option that could further facilitate the prevalent use of these heavy computational modules for less advanced users. So in general, I am positive with regard to the degree of novelty of this effort, the study design and to the most part, I believe the arguments presented are providing convincing ground in supporting the suitability of the software for pertinent analysis. Overall, I am not having huge concerns, however due to the complexity of each of these modules and inevitable challenges to concert them into a pipeline, here are my comments and questions I wish to raise with hope of enhancing and strengthening this study.

Please find below first my general comments regarding the main modules of the manuscript, proceeded with more granular ones with reference to the presentation flow of the paper, and finally closing remarks toward the end.

Registration:

Considering the heavy computational dimension of this step, I think it is essential to inform the users about this aspect in terms of the run time and memory needed for the sample of 'x' size. Naturally this would be a function of the number of the tiles and their respective sizes which is not properly discussed. The authors talk about the 10% overlapping margin but what would be the downstream effect of different settings of this overlap percentage?

Regarding the tiles, what are their actual size, how this size is resolved, and what is its effect on the registration quality for example? Furthermore, if the tile size is adaptive, the arguments regarding the downstream effect of different choices on performance of the modules is an essential add. My understanding suggests that tending to smaller sized tiles could conceivably lead to better registration, but does it have any costs in terms of the number of cells being left out due to tiling, overlapping, and stitching algorithm? In the current form, are we having any statistics regarding the adverse effect of the tiling as the cells being spared from the analysis?

In cases where microscope has stitched the image, is tiling and restitching of MARQO posing any multiplicative error? And in cases that tiled images are readily available as the microscope output, is it possible to adopt them as the considered tiles here? What would be the computational aspects around these different scenarios? In general, I think more arguments regarding these issues is highly appropriate.

Segmentation:

In general, I think more reasoning about the real need of the new segmentation method is needed. Here authors are referring to the composite segmentation method, however they are providing their benchmark against the STARDIST as adopted through QuPath. The linear correlations are not sufficient if the authors are attempting to establish their method as the novel one, as in such case, one would expect working with more standardized datasets, gold standard masks, and provide a thorough comparison with other methods and obtain the confusion matrix and other formal benchmarking indices like F1 score to support their method [1].

If the method is not that novel, then the question is what are we gaining by tiling, and using the STARDIST and voting scheme as compared to the use of the STARDIST straight on the whole image and first layer of the stained image? Wouldn't it free us from one extra step? Shouldn't the voting scheme be used as a quality control measure of the consecutive staining than a tool in deciphering the cells? I think these points need a bit more clarification.

Also I would argue that the correlation plots generated, are suggesting the effort of composite segmentation in imitating the STARDIST not its superiority! This needs to be established with more supervised approach where the true annotation of the cells on the image is available and benchmarking indices could be obtained with the independent use of MARQO for detection of the cells.

Still within the space of what is presented, based on the tiles and semantic nuclear segmentation of STARDIST and voting criteria of 60% over different stains within 3 microns, the cells are being segmented. So, if this is done after the registration, 3 micron seems to be rather high and also 60% seems to be on the low side! I welcome authors to provide more reasoning on why they think these two parameters are optimal choice to be set as default. I particularly think it is very good idea to visually show the optimality of these two parameters with a graph represented over a range of the 'vicinity of the cell detection' parameter (e.g. 1-5 micron on x) and 'positive voting' percentile (e.g. 40-90% on y) with an objective function illustrating the

behavior of that function and the derived optimal parameters mapped to the function's extremum. This should be immensely informative also for the users (considering specially they can set their custom values) to choose a proper combination with respect to that two-dimensional survey.

Also as mentioned in registration previously, this step is also dependent on the tiles and their size, so the similar arguments regarding the effect of the tile size in efficiency of the segmentation is prevailing here as well. Also the computational aspect of this step with regard to the number of the tiles would be relevant to discuss.

Quantification and Analysis:

The quantification is based on the artificial dilation of the nucleus mask. While this is quite mainstream, have authors considered the use of the cytoplasmic markers (in case they are present in the maker panel) to enhance their masks? While this is discussed in the discussion for future development, considering the importance of this issue, authors could consider already integrating an early version here.

Also, the two unsupervised methods used are appealing, and I suggest having a plot maybe in supplementary exhibiting the direct output of their use for an example input. Also, authors are discussing the unsupervised clustering of the cells and supervised clustering of the markers (L173) which is very confusing as I assume the markers intensity inevitably should be used for cell clustering and vice versa? In general, the choice of K-means and GMM is rather unsupported in the text. One could consider multiple other methods such as NMF (for decomposing the sample and feature space simultaneously), Gamma mixture model, ICA, etc.

Minor:

The concept of the semi-supervised and automated are not clearly defined. Specially I think the semi-supervised is not a suitable term to be used here as this concept is commonly reserved in ML space for the cases where the dependent variable outcomes are not fully labeled. Considering the couple of times mentioning of 'machine learning' in the paper, it was a bit confusing to adopt the new concept the authors try to convey which in fact is the reclustering capability of cell types by users (user-mediated clustering, hybrid, semi-automative?).

The other one is the 'automated', for example mentioned on the L62-64 as desirable but again L80 refers to the issue arising from automated signal detection. I recommend authors either adopt alternative terms or be more clear about each of these and establish them as keywords for their specific use.

I also think would be possible to talk more about the advanced ML models used in this space for example on cell detection [3-4] and in general providing more arguments supporting the development of this software as mentioned in L60-62.

L119, 368-370: It's great to have specially the affine registration for the alignment of the tiles and the program being able to forgo a registration step if it is not needed, however due to the computational cost of this transformation, does the program have automatic detection and bypass of this step in case this is not needed. And considering the parameters being optimized on MICSSS images, how transferrable the results would be to the other modalities. If they are tweakable, how users are made aware to choose the proper values for their respective modalities.

L124 talks about the benchmarking data outputs generated. I think it would be great to have some of the most pivotal ones reflected in a graph as a supplementary figure.

L128: It's suggestable to clearly explain the composite segmentation herein with reference to its underlying STARDIST core method upfront.

L145: Can you provide your insight regarding the elimination of the red blood cells in terms of sensitivity and specificity rates?

L151: The semi-manual segmentation comparison seems a bit non-specific. In order to avoid confounding errors of two independent modes and gain more orthogonal insight, wouldn't it be more meaningful to have the comparison with fully automated segmentation and perfectly manual ones separately?

L161: Why not comparing the outputs with at least one other mainstream method like Cellpose, specially so that the underlying method for MARQO is based on STARDIST?

L174: You may consider using different terminology.

L187-189: Is the manual counting of the positive cells performed on the raw image or the one after algorithmic cell detection is established?

L214: What is the motivation for choosing these two markers?

L240-242: I recommend providing a preliminary figure in Supplement showing the result of your testing for each modality.

L243: Referring to the high accuracy here is not quite convincing only based on the r-square and p-values. Why not considering the formal comparison of the MARQO performance with confusion matrix and its derived metrics such as F1 score?

L241: CyCIF

L268: ...tend 'to' be ...

L438: It is unclear where these 60 GMMS are coming from!

Fig3: In general couple of points were not clear with these graphs that I think could be clarified. -how the random graphs are obtained? -what was the criteria for choosing the ROIs? -what are the dashed line in panel E?

Fig4: B: Where these boxes close up images are placed on the tissue? C: Are the points representing the tiles? B: Typically the CD3 and PanCK are orthogonal, is that the reason why authors used these two here? Despite this fact, the right side of the 4th row is exhibiting the hump with a margin marked positive for both markers. How this is explained and in general, situations like these are calling for a more robust quantifications such as F1 score than linearity assessment emphasized throughout.

Fig5: I recommend a more versatile comparison both in terms of markers and methods (other than QuPath), specially with reference to the ground truth for settled cell phenotypes. Like in panel B, PanCK, it seems there are at least two quite distinct cells to the left side of the image with positive signal that are not detected!

Fig6: A: What are the x-axis of this panel and are those meant to be vertically parallel? If they are the patients, why the one on the top is missing two and what is the order from the first heat map to the one below? B and C: Are these two lesions on the slide being treated differently? In general, depending on the analysis, the lesions on the same slide may (e.g. clearly in cases of neighborhood analysis) or may not (e.g. threshold gating) be treated independently. Would be good if authors could appreciate this issue specially regarding how MARQO will perform its analysis.

FigS1A: What would be the effect of having "overlapping borders tiles removal" after application of vector field alignment?

FigS3: The quality of the image for FOXP3 is low and I was not able to discern the detections. However, for the PanCK, there are too many greens (and almost no blue) suggesting the high rate of false positive?

SupTab1: CyCIF

SupTab2: Maybe removing the almost identical fourth sub-table by adding the PDL-1** to the end of the third sub-table as the 10th row, amending the title of the 3rd sub-table accordingly, and adding the note **: Only applicable to the MICSSS on whole-slide NSCLC resections.

The other overall issues that I believe are meaningful for the authors to consider is

- the more versatile use of the markers as specially they are readily available throughout their study, specially immune ones like CD19/CD20 in last figure to detect possible community of the B cells suggestive of the tertiary lymphoid structures and maybe an additive figure as the pseudo-image of the exemplar tissue with the T can B cells overlaid.

- discussing the optimal performance envelope of the software in general and specifically, with regard to patients' variability and best practices to diminish its adverse effect toward optimal take aways of the pipeline -something that is completely left out in the use case of the software reflected on the last figure.

- providing more arguments around the scalability and computational edges of this tool with regards to the big cohort studies possibly involving multiple imaging modalities and containing dozens to hundreds of samples.

- placing their findings in the context of newly introduced and powerful transformer-based methods for deciphering the mesoscale architecture of the tissue [4-5].

Lastly, as presented modules are exhibiting a natural ordinality, the authors may wish to consider, at their discretion, an acronym more reflective of this fact such as MIRSQA (Multiplex-imaging Integrative Registration Segmentation Quantification and Analysis) or alike!

Sincerely,
Ali Amiryousefi

Bibliography

- [1] Ma, J., Xie, R., Ayyadhury, S., Ge, C., Gupta, A., Gupta, R., ... & Wang, B. (2024). The multimodality cell segmentation challenge: toward universal solutions. *Nature methods*, 1-11.
- [2] Spitzer, H., Berry, S., Donoghoe, M., Pelkmans, L., & Theis, F. J. (2023). Learning consistent subcellular landmarks to quantify changes in multiplexed protein maps. *Nature Methods*, 1-12.
- [3] Liu, C. C., Greenwald, N. F., Kong, A., McCaffrey, E. F., Leow, K. X., Mrdjen, D., ... & Angelo, M. (2023). Robust phenotyping of highly multiplexed tissue imaging data using pixel-level clustering. *Nature Communications*, 14(1), 4618.
- [4] Xu, H., Usuyama, N., Bagga, J., Zhang, S., Rao, R., Naumann, T., ... & Poon, H. (2024). A whole-slide foundation model for digital pathology from real-world data. *Nature*, 1-8.
- [5] Chen, R. J., Ding, T., Lu, M. Y., Williamson, D. F., Jaume, G., Song, A. H., ... & Mahmood, F. (2024). Towards a general-purpose foundation model for computational pathology. *Nature Medicine*, 30(3), 850-862.

Remarks on code availability:

As the main offer of this study, I checked the GitHub repository, which is quite organized, with adequate instructions for installing the software with Docker. The README page is quite comprehensive with clear guidelines and step by step installation tips. The software interface was running smoothly without any particular issue. The data used in the manuscript is also provided. Running couple of examples, I see the conformity with the supplementary videos and obtaining outputs resembling the presented results in the paper.

I have not fully checked all the underlying code of the directories, but couple sampling and overall skimming suggested an organized and professional coding scheme throughout the full repository.

Reviewer #3 (Report for the authors (Required)):

This manuscript by Backup et al describes an image alignment and analytic tool for multiplex immunohistochemical assays, which they term MARQO. The software is designed to handle a diversity of multiplex imaging approaches/platforms, including MICSSS. Ultimately all of the data input types are variation on theme - images of different targets obtained from the same sample - in some cases and image stack and in some cases, repetitive single-plex IHC.

The manuscript is clearly written, with well developed figures. The manuscript outlines the general capacities of the software package well. The authors compare the software to QuPath, Halo and Visiopharm, but do not reference other software packages that may be available - both commercial and research developed. The overall intended use of the software is analysis of a diversity of tissue inputs, including whole sections, biopsies and tissue microarrays with the analytic goal generally focused on immuno-oncology.

The workflow, image input and output parameters are well described and common to many image processing platforms. To describe the package in lacking in features would be unfair, however the software is limited in the tool set for which it can segment cells and offers no tools for extracellular matrix evaluation or other complex tools. The tools used for cell segmentation are reasonable, but are likely insufficient for some complex multi-cell type analytic settings.

The challenge is evaluation of the package. The primary argument the authors take is that MARQO is something that can be implemented, works on multiple multiplex datasets and is optimized for MICSSS. It is unclear how widely MICSSS is used. Although the authors provide some data comparing MARQO to the previously mentioned software package, they fail to demonstrate that MARQO has a clear benefit. The measure of benefit in image processing is generally performing an analysis that has not been possible previously, or demonstrating a outcome that can not be defined by other software, in other words superior results.

The software is good, but it fails to generate the enthusiasm to why a researcher would change pipelines if they are already analyzing data. The investigators discuss the benefits of local compute, however anyone doing these studies hopefully not limited by compute. Ultimately the real issue is total time - both human at computer and computer compute required to generate a usable data package. The measure of usability is a critical and commonly missing element of any evaluation of data pipelines, and as any experienced multiplex researcher, the process can take days to weeks to manage just one experiment.

Remarks on code availability:

Although I like the idea of being able to review the code, many reviewers work in highly secure IT environments, and the capacity to download and review the code is not a viable approach from an IT security perspective.

Version 1:

Decision Letter:

Dear Dr Gnjatic,

Thank you for your revised manuscript, "MARQO pipeline resolves multiparametric cellular and spatial organization in cancer tissue lesions". Having consulted with the previous reviewers, I am pleased to write that we shall be happy to publish the manuscript in *Nature Biomedical Engineering*. I am very sorry for the unusually long delay.

We will be performing detailed checks on your manuscript, and in due course will send you a checklist detailing our editorial and formatting requirements. You will need to follow these instructions before you upload the final manuscript files.

Best wishes,

Barbara Cheifet

Editor
Nature Biomedical Engineering

Reviewer #1 (Report for the authors (Required)):

Dear authors,

thanks for submitting a revised version of the manuscript.

All my major and minor comments have been properly addressed so that I can fully recommend the publication of the article. Thanks for developing and disseminating a software package that is addressing a timely and relevant shortcoming in the software landscape for the analysis of large, multiplexed image data.

Best regards

Reviewer #1 (Remarks on code availability):

The code is well documented following state of the art community procedures.

Reviewer #2 (Report for the authors (Required)):

Dear Authors,

Thank you for providing your responses to my comments and updating the manuscript accordingly. I believe the manuscript is now more cohesive and conductive of your added value. Up to the quality of some of the supplementary figures -which I believe would be handled by you and the publishing house- I have no further comments.

Thank you,
Ali Amiryousefi

Reviewer #2 (Remarks on code availability):

As the main offer of this study, I checked the GitHub repository, which is quite organized, with adequate instruction for installing the software with Docker. The README page is quite comprehensive with clear guidelines and step by step installation tips. The software interface was running smoothly without any particular issue and the data led to the produced results in the paper is also provided. Running couple of examples, I see the conformity with the supplementary videos and obtaining outputs resembling the presented results in the paper.

I have not fully checked all the underlying code of the directories, but couple sampling and overall skimming suggested an organized and professional coding scheme throughout the full repository.

Version 2:

Decision Letter:

Dear Dr Gnjjatic,

I am happy to inform you that your manuscript, "Multiparametric cellular and spatial organization in cancer tissue lesions with a streamlined pipeline", has now been accepted for publication in *Nature Biomedical Engineering*.

Over the next few weeks, the figures will be checked for production quality, the text edited to ensure that it conforms to house style, and the manuscript typeset.

Our Articles are published about 40 days after the acceptance date (we recommend that you inform your institutional press office of this timeframe), and you will be notified of the actual publication date a few days in advance. Articles can be published any working day of the week, and are pushed live shortly after 10 am London time.

Publishing agreement. You will be asked to digitally sign a publishing agreement (grant of rights). After the signed

publishing agreement has been received, the proofs of the article will be sent to you for review. If you have any queries during the production process, or you cannot meet the requested deadline for returning the proofs, please contact rjsproduction@springernature.com.

Nature Biomedical Engineering is a Transformative Journal. Authors may publish their research with us through the traditional subscription access route, or make their paper immediately open access through payment of an article-processing charge. More [information about publication options](https://www.springernature.com/gp/open-research/transformative-journals) is available.

You may need to take specific actions to [comply](https://www.springernature.com/gp/open-research/funding/policy-compliance-faqs) with funder and institutional open-access mandates. If the work described in the accepted manuscript is supported by a funder that requires immediate open access (as outlined, for example, by [Plan S](https://www.springernature.com/gp/open-research/plan-s-compliance)) and your manuscript was originally submitted on or after January 1st 2021, then you should select the gold OA route. Authors selecting subscription publication will need to accept our standard licensing terms (including our [self-archiving policies](https://www.springernature.com/gp/open-research/policies/journal-policies)), and these will supersede any other terms that the author or any third party may assert apply to any version of the manuscript.

Acceptance of your manuscript is conditional on agreement, by all authors, with both our [media embargo](http://www.nature.com/authors/policies/embargo.html) and [confidentiality and pre-publicity](http://www.nature.com/authors/policies/confidentiality.html) policies. In particular, you may arrange your own publicity of the Article (for instance, through your institutional press office), as long as you ensure that journalists strictly adhere to the media embargo.

To assist you in disseminating the work, as soon as the Article is published you will be able to take advantage of the Springer Nature [SharedIt](https://www.springernature.com/gp/researchers/sharedit) initiative to [generate a unique shareable link to the Article](http://authors.springernature.com/share) that will allow anyone (with or without a subscription) to read it. Recipients of the link who are subscribers will also be able to download and print the PDF.

Thank you for having submitted this work to *Nature Biomedical Engineering*.

Best wishes,

Barbara Cheifet
Editor
Nature Biomedical Engineering

Point by point

Dear Dr Gnjatic,

Thank you again for submitting to *Nature Biomedical Engineering* your manuscript, "MARQO pipeline resolves multiparametric cellular and spatial organization in cancer tissue lesions". The manuscript has been seen by three experts, whose reports you will find at the end of this message.

You will see that the reviewers appreciate the work. However, they express concerns about the degree of benchmarking provided and the wider utility of the pipeline (such as its extensibility), and provide useful suggestions for improvement. We hope that with substantial further work you can address the criticisms and convince the reviewers of the merits of the study.

When you are ready to resubmit your manuscript, please upload the revised files, a point-by-point rebuttal to the comments from all reviewers, the reporting summary, and a cover letter that explains the main improvements included in the revision and responds to any points highlighted in this decision.

Please follow the following recommendations:

- * Clearly highlight any amendments to the text and figures to help the reviewers and editors find and understand the changes (yet keep in mind that excessive marking can hinder readability).
- * If you and your co-authors disagree with a criticism, provide the arguments to the reviewer (optionally, indicate the relevant points in the cover letter).
- * If a criticism or suggestion is not addressed, please indicate so in the rebuttal to the reviewer comments and explain the reason(s).
- * Consider including responses to any criticisms raised by more than one reviewer at the beginning of the rebuttal, in a section addressed to all reviewers.
- * The rebuttal should include the reviewer comments in point-by-point format (please note that we provide all reviewers with the reports as they appear at the end of this message).
- * Provide the rebuttal to the reviewer comments and the cover letter as separate files.

We expect that you will be able to resubmit the manuscript within 16 weeks of receiving this message. If this is the case, you will be protected against potential scooping. Otherwise, we will be happy to consider a revised manuscript as long as the significance of the work is not compromised by work published elsewhere or accepted for publication at *Nature Biomedical Engineering*.

We hope that you will find the referee reports helpful when revising the work. Please do not hesitate to contact me should you have any questions.

Best wishes,

Pep

We thank the reviewers for their feedback. Please find our responses to the comments (black) in blue. We strongly believe all comments have helped to significantly strengthen the impact and clarity of the article, for which we are grateful to the reviewers and editor.

Reviewer #1 (Report for the authors (Required)):

The manuscript 'MARQO pipeline resolves multiparametric cellular and spatial organization in cancer tissue lesions' by Mark Buckup et al. addresses the need to analyse imaging data from tissue proteomic assays. In particular, it aims to provide a single platform to cope with various assay protocols and avoid the cumbersome, lengthy, and error-prone habit of using different software packages sequentially. As a use case, the authors demonstrate CD8 T cell enrichment in hepatocellular carcinoma responders to neoadjuvant cemiplimab in a phase II clinical trial.

The workflow and use case are presented clearly and understandably by following well-accepted practices. In particular, the software's documentation for that workflow is beyond average and demonstrates the authors' wish to provide a tool that the community picks up and uses.

The presented workflow is based on well-accepted and known algorithms and aims to facilitate the use and application of known tools rather than the development of new tools. However, the availability of open-source workflows with low entry barriers for spatial proteomics is currently a primary challenge in facilitating a broader usage of these technologies. Thus, the presented workflow is timely and fits the community's needs.

It would benefit the manuscript if the authors outlined how to disseminate the workflow and ensure that it is used and further developed by the community. Also, a strategy for adapting and implementing new protocols and image analysis routines is missing. However, this is critical to guaranteeing the sustainability and longevity of that workflow.

Please find my detailed comments below:

Comment 1

Major :

I. 355 ff. You claim that it is an advantage that the ML algorithms you provide do not need to be retrained. In practice, it turned out that retraining is necessary in some instances. Thus, the workflow design limits the possibility of adapting to such cases. This might render the workflow less valuable in the future. The ability to retrain ML algorithms (e.g. Stardist) should be made possible as an option.

We thank the reviewer for this comment. We have used the StarDist segmentation package, which is a pre-trained package that was imported into our environment for MARQO. Because this package has tunable parameters to adjust its sensitivity, we have made these parameters customizable in the launch application. In addition, we have added an additional segmentation package called CellPose, which the

user may choose to use instead of our preexisting StarDist singular or composite segmentation methods. Parameters affecting its performance, along with descriptions of what each parameter does, have also been added to our application. We have added a short description and citation of this new segmentation model in lines 454-456.

Comment 2

I. 159 ff. The cited literature does not support the claim that a dice score of 0.85 indicates an automated segmentation better than a manual one. It would help if you were careful not to overinterpret your findings.

We thank the reviewer for this comment. Because we did not have true negatives when segmenting cells, we used DICE scores, but we agree that a 0.85 DICE score of MARQO to the semi-manual segmentation performed in QuPath does not imply superiority over a manually-derived segmentation by a pathologist. We have therefore asked our pathologist to manually segment true cells in 15 ROIs, with a specific methodology defined in our methods section (see lines 616-660). While Figure 2A focuses on composite segmentation improvements when compared to singular stain segmentation, 2B focuses on MARQO's performance in different densities, and our newly added 2C and 2D compare MARQO's composite segmentation to the manual quantification of our pathologist. We have also rephrased our statement so that a 0.83 DICE score means the model performs similarly to the manual work done by a pathologist, despite having advantages to the singular-stain segmentation method, which is a conventionally used methodology in the field. Kindly see revised supplementary Figure S3D for our DICE calculations for all 15 files (see lines 179-184).

Comment 3

L 167 ff. It is well-known and accepted that ML introduces a bias in any image analysis. However, it is questionable that a workflow with parameters set by a user is introducing less bias as claimed for the presented workflow. The manuscript will significantly profit from a fairer and more evidence-based discussion.

We thank the reviewer for this comment. While we tried to emphasize the lack of training for diverse datasets in MARQO, we agree that the pipeline will have similar bias to ML algorithms with predefined variables. Therefore, we extracted the statement suggesting bias only in non-MARQO supervised algorithms and now have emphasized MARQO's ability to process multiple data types without extra training, as well as possible customization with tunable parameters that are added directly to the launch application. Please refer to lines 190-196 for these alterations to the text.

Comment 4

I. 286 ff. The 'semi-supervised' approach presented in your manuscript is useful and provides insightful results. However, it comes with the expense of needing a well-trained expert to run and evaluate the workflow. This aspect is missing in the discussion section.

We thank the reviewer for this comment. Like other platforms, MARQO is not yet fully unsupervised and requires a pathology expert to assess this last step in the methodology when classifying cell types.

However, we have tried to alleviate the workload for the expert by giving MARQO the ability to perform this step in parallel for multiple samples. We have updated lines 329-333 to reflect this feedback.

Comment 5

Minor :

I. 68 The ability to run the analysis in the cloud is highly appreciated. However, pointing out legal issues with patient samples being analysed in the cloud might be necessary to address, at least in the conclusion/discussion section.

We thank the reviewer for this comment. We have now added the consent details in the Materials and Methods section, clarifying that it includes cluster storage of the data. We have also removed all mentions of "cloud" and replaced them with "cluster," to further remove any ambiguity with shareable cloud storage. If further clarification is required, we are happy to include a brief statement in the Discussion as well.

Comment 6

I. 87 Comparing a tailored workflow with a generic tool like QuPath seems a bit unfair unless you specifically point out which steps will not be possible in QuPath.

We thank the reviewer for this comment. While we discuss the main aims and advantages of MARQO to current image analysis tools in the previous paragraph, we did not discuss the specific limitations of QuPath when analyzing our multiplex technology, MICSSS. Please see updated lines 93-99, where we specifically discuss why we were unable to process MICSSS in the versatile QuPath software package, then segueing into how MARQO allowed us to make observations of spatial immune responses to ICB in cancer using MICSSS.

Comment 7

I. 130 Please provide more details on the denoising algorithm and why it is necessary.

We thank the reviewer for this comment and agree that the use of "denoise" was inaccurately used to describe that repetitive nuclear staining clarifies cell boundaries. By iteratively segmenting the same cell across stains, some rounds having poorer or stronger staining than others, a composite mask can be created that more accurately describes the true boundaries of cells in the region of interest. We hope this message is now articulated well in the altered text, lines 138-140.

Comment 8

I. 140 The decision process when a nuclear mask is rejected is outlined in detail in the MM section. It might be helpful to mention it in the main text.

We thank the reviewer for this comment. We have swapped the explanation of the in-detail methods with the explanation of the main text so that the main text now contains details on the decision-making process regarding whether or not a nuclear mask is rejected. Please refer to lines 153-161 in the main text and 443-448 in the methods for these changes.

Comment 9

I. 197 ff. In this application, the workflow is as accurate as the pathologist's manual annotation. However, there might be cases that serve as counter-examples.

We thank the reviewer for this comment. While we had difficulty finding examples where automated annotations would outperform a pathologist, such as high-volume practices or extremely difficult differential diagnoses where unbiased approaches may have benefit, we instead included this point in our discussion. Please see lines 323-344, where we discuss the difficulty of automatically recognizing technical issues (such as border staining or blurry areas), which the trained eye may catch, as well as the converse, where the unbiased algorithm may outperform the expert. In tandem with comment 4, we also discuss how we are at a transitional stage in digital pathology, and how AI-decision making tools will likely replace our current semisupervision techniques in the future.

Comment 10

I. 297 ff. It might be worth commenting on which of the parameters needed to be fine-tuned to adapt to the different assays and how much expert knowledge is needed to find the right parameters.

We thank the reviewer for this comment. To complement our response to comment 3, we have created a parameters markdown section in our application, with thorough descriptions per parameter added. In revised text lines 193-199, we have provided this information as well as our recommendation not to change the optimized values unless the user has adequate experience in image analysis. We have provided the information below as well:

StarDist parameters:

- Pixel Size (**Expertise Needed - Low**): Defines the physical size of each pixel in the image (e.g., microns per pixel). **When to adjust? Rarely.** The pipeline default values will re-scale the input images to match the pixel size of the images used for training the Stardist predefined models.
- Probability Threshold: (**Expertise Needed - Moderate**): Minimum probability to detect objects. Typical values range from 0.25 - 0.90. **When to adjust?** When false positives or missed detections are frequent.
- NMS (Non-maximum suppression) Threshold: (**Expertise Needed - Moderate**): Controls how close two detected objects can be before being merged into one. Lower values are stricter. Typical values range from 0.3 to 0.7. **When to adjust?** When objects are merged incorrectly (e.g., touching cells).

Cellpose parameters:

- Flow (**Expertise Needed - Moderate**): Determines the flow threshold, which controls how strict the model is when identifying the flow direction of each pixel. Higher values may detect more objects but increase false positives. **When to adjust?** If artifacts are misclassified as cells.
- Cell Diameter (**Expertise Needed - Low**): The approximate size (diameter) of the cells in pixels. Default value of 0.0 allows the model to calculate the average size of cells. **When to adjust? Rarely.** If the objects are much larger or smaller than expected.

General Recommendations:

Model	When to Change Parameters	What to Watch For
StarDist	Adjust NMS threshold (Non-maximum suppression) if objects are too close and merging, or adjust Probability threshold if you want to capture weaker	If too many false positives, increase Probability threshold . If nearby objects are being

	signals.	merged, lower NMS threshold .
Cellpose	Adjust Cell Diameter if the cells are much larger or smaller than typical. Adjust Flow if cell shapes are very irregular or if you are targeting a specific cell population.	If false positives occur, increase Flow . If large objects are missed, increase the Cell Diameter .

Comment 11

I. 452 Ideally, the data will be deposited in a public repository.

We thank the reviewer for this comment. We have anonymized patient information and organized cell type and co-expression data into a downloadable excel file, which we have provided as a supplementary file, named Extended Data Table 1.

Remarks on code availability:

The code is sufficiently documented and contains instruction on how to install and run the software.

Reviewer #2 (Report for the authors (Required)):

Buckup. M. et.al. introduces MARQO; a computational pipeline to streamline the workflow of multiple modules necessary for extracting higher level insights from the cancer tissue lesions over different multiplex immunostaining-based imaging modalities. Namely, each of these modules as registration, segmentation, and quantification, are deeply intertwined in terms of their parameters and outputs and are pivotal parts of multiplexed immunohistochemistry imaging studies. As such, they have been quite heavily studied and matured to varying degrees. The bold offering as the integration of all these modules topped with an even more preferential yet necessary analytical one as the main thesis of this study, is a meaningful step toward more efficient knowledge elucidation of the underlying images. Also providing the possibility for the users to interact and influence the output of these modules through a graphical user interface is a great option that could further facilitate the prevalent use of these heavy computational modules for less advanced users. So in general, I am positive with regard to the degree of novelty of this effort, the study design and to the most part, I believe the arguments presented are providing convincing ground in supporting the suitability of the software for pertinent analysis. Overall, I am not having huge concerns, however due to the complexity of each of these modules and inevitable challenges to concert them into a pipeline, here are my comments and questions I wish to raise with hope of enhancing and strengthening this study.

Please find below first my general comments regarding the main modules of the manuscript, proceeded with more granular ones with reference to the presentation flow of the paper, and finally closing remarks toward the end.

Comment 12

Registration:

Considering the heavy computational dimension of this step, I think it is essential to inform the users about this aspect in terms of the run time and memory needed for the sample of 'x' size. Naturally this would be a function of the number of the tiles and their respective sizes which is not properly discussed. The authors talk about the 10% overlapping margin but what would be the downstream effect of different settings of this overlap percentage?

We thank the reviewer for this comment. We have revised the text (lines 394-397) to rationalize the default value of 1000 by 1000 pixels for our tiles, with an added 10% overlap in the registration step. It is, in fact, a trade-off between smaller but many more tiles to process versus fewer and larger tiles that require increased computational resources. To address this comment, we have updated our Figure S2 to include subfigure C, depicting the run times for a 1000 by 1000 pixel region of interest. On the left, the region is divided into sixteen 500x500pi tiles and each tile is run sequentially. In the middle, four 1000x1000pi tiles are run sequentially. On the right, one 2000x2000pi tile is run. This is all done on one CPU containing 6GB of memory. Because of these run times for running an identical amount of tissue area, we choose 1000x1000pi as our default tile size. In addition, to provide the user with information regarding computational times for specific tasks in the methodology, we refer the reviewer to comment 50. Lastly, we optimized 10% overlap during the registration step by having our pathologist review the final stitched images (post-registration) and assess if the stitching was good or poor. Ideally, we wanted to decrease this value as much as possible, contingent on adequate stitching deemed by our pathologist, so that we would not waste computational time and resources on redundant areas between adjacent tiles. We settled on 10%, which is in agreement with other literature during the tiling and stitching process, such as the ASHLAR pipeline (citation #12). Please see revised lines 425-427 for this change.

Comment 13

Regarding the tiles, what are their actual size, how this size is resolved, and what is its effect on the registration quality for example? Furthermore, if the tile size is adaptive, the arguments regarding the downstream effect of different choices on performance of the modules is an essential add. My understanding suggests that tending to smaller sized tiles could conceivably lead to better registration, but does it have any costs in terms of the number of cells being left out due to tiling, overlapping, and stitching algorithm? In the current form, are we having any statistics regarding the adverse effect of the tiling as the cells being spared from the analysis?

We thank the reviewer for this comment and kindly ask to refer to comment 12, where we have provided a thorough reasoning on the size of tiles and their computational resource needs. In short, we optimized tile size to be 1000 by 1000 pixels, which translates to approximately 500 by 500 microns at the conventional 20x scanning setting. This parameter is not tunable, as the much of the downstream analysis would significantly change due to the greater computational demands. In regards to the registration step, we add a 10% overlap to every tile. In other words, a tile that is by default 1000x1000 pixels becomes 1100x1100 pixels; its surrounding tiles also get extended. Therefore, there is a redundant tissue area in adjacent tiles. We do this so that after an elastic registration, if tissue was stretched, say, towards the inside of the tile, instead of there being empty space at the boundary of the tile, that area is now populated with the extra 10% tissue area. In this situation, tissue area that was outside the bounds of the original tile (likely due to tissue damage and warping over sequential stainings) is returned to the bounds of the tile. After registration, we eliminate the 10% overlap so that all tiles are unique and contain non-redundant tissue area. 10% was optimized as explained in comment 12. With this rationale, the image can be stitched back to its original size without any loss of cells.

Comment 14

In cases where microscope has stitched the image, is tiling and restitching of MARQO posing any multiplicative error? And in cases that tiled images are readily available as the microscope output, is it possible to adopt them as the considered tiles here? What would be the computational aspects around these different scenarios? In general, I think more arguments regarding these issues is highly appropriate.

We thank the reviewer for this query. Large image inputs that were stitched together by the microscope (in varied ways depending on which microscope) have inconsistent ways of denoting tile indices and spatial relations, if at all, which is further confounded and made impossible to use when performing multiplex staining. For example, a tile in the top left of the first image may reside in the center of the tissue during the first staining, whereas the same top left tile on a sequential staining may then correspond to an entirely different part of the tissue because of tissue sliding, tissue damage, off centered microscope calibration, and other limitations to consistency in multiplex imaging technologies. We have not yet encountered or addressed the specific situation in which the microscope outputs precise tile information and the technology used does not shift across multiplex staining (ex. CODEX).

We tested multiple image types and technologies on the MARQO pipeline and have successfully produced results, including the multipyramidal .svs and .ndpi images that are often outputted when scanning an IHC assay. While these pyramidal image arrays contain lower-resolution whole-slide images, MARQO pulls the highest-resolution image and splits the image into 1000x1000 pixel tiles, in order to keep the highest resolution for the most optimal analysis. The tiling step is also essential since loading a massive image array (some up to GB's of data) remains difficult to open, adapt, and visualize with the

typical RAM and computational resources available to a user. Please see comment 12 for the rationale behind the tile dimensions we chose.

Lastly, we consider it advantageous that MARQO performs its own, independent tiling. MARQO provides another tool to split a large image into multiple smaller images. In addition, because MARQO tiles the larger image input into smaller pieces and permits the user to optionally re-create the larger final stitched image, MARQO remains independent of the quality and variability in the microscope's stitching abilities.

Comment 15

Segmentation:

In general, I think more reasoning about the real need of the new segmentation method is needed. Here authors are referring to the composite segmentation method, however they are providing their benchmark against the STARDIST as adopted through QuPath. The linear correlations are not sufficient if the authors are attempting to establish their method as the novel one, as in such case, one would expect working with more standardized datasets, gold standard masks, and provide a thorough comparison with other methods and obtain the confusion matrix and other formal benchmarking indices like F1 score to support their method [1].

If the method is not that novel, then the question is what are we gaining by tiling, and using the STARDIST and voting scheme as compared to the use of the STARDIST straight on the whole image and first layer of the stained image? Wouldn't it free us from one extra step? Shouldn't the voting scheme be used as a quality control measure of the consecutive staining than a tool in deciphering the cells? I think these points need a bit more clarification.

We appreciate the reviewer's thoughtful insight on how to strengthen our segmentation model discussion and comparisons. We agree that having one conventionally used segmentation method and comparing it to its analogous counterpart in QuPath, despite being semi-manually assessed by our pathologist, poses a weakness in our manuscript. To address comments 1, 2, 16, 28, 29, as well as this one, we have 1) compared our novel composite StarDist segmentation model to a ground truth of 15 ROIs with cells manually segmented by our pathologist and 2) added an entirely new segmentation package that the user can choose to use instead of the composite StarDist package. For more details, please see comments 1 and 2. To address the other specific points raised by the reviewer, we emphasize that MARQO is a package that includes many steps in one; the tiling is a scaffold to make the registration possible, as well as permit parallel computing across multiple cores with small computational demands each. Therefore, we also perform the composite segmentation per tile, as doing it for the whole image would require a much larger computational demand. In addition, the elastic registration must precede segmentation as it improves the accuracy of our segmentation model. Lastly, it is not our goal to benchmark StarDist -- in fact, we have made the parameters customizable directly in the user-operated application. Rather, the goal of a tile-based composite segmentation is to remove artifacts that were deemed as cells within a singular segmentation, as we show in Figure 2A. We have revised our language in lines 179-184 to not imply that the composite segmentation is superior to its singular counterpart, but rather that it removes artifacts and improves cell boundaries more than the singular segmentation.

Comment 16

Also I would argue that the correlation plots generated, are suggesting the effort of composite segmentation in imitating the STARDIST not its superiority! This needs to be established with more

supervised approach where the true annotation of the cells on the image is available and benchmarking indices could be obtained with the independent use of MARQO for detection of the cells.

We thank the reviewer for this comment. Since the other reviewers raised a similar concern, we kindly ask them to refer to comments 2 and 15, where we discuss our new validation method for our segmentation.

Comment 17

Still within the space of what is presented, based on the tiles and semantic nuclear segmentation of STARDIST and voting criteria of 60% over different stains within 3 microns, the cells are being segmented. So, if this is done after the registration, 3 micron seems to be rather high and also 60% seems to be on the low side! I welcome authors to provide more reasoning on why they think these two parameters are optimal choice to be set as default. I particularly think it is very good idea to visually show the optimality of these two parameters with a graph represented over a range of the 'vicinity of the cell detection' parameter (e.g. 1-5 micron on x) and 'positive voting' percentile (e.g. 40-90% on y) with an objective function illustrating the behavior of that function and the derived optimal parameters mapped to the function's extremum. This should be immensely informative also for the users (considering specially they can set their custom values) to choose a proper combination with respect to that two-dimensional survey.

We thank the reviewer for this comment and agree that these values seem rather arbitrarily chosen without much rationale in the text. The MARQO pipeline consists of many packages and steps, with each having its own set of parameters that were either optimized by third parties, optimized by us, or not yet entirely optimized but work with great results in our application. Rather than explain the optimization behind every variable, we have taken many of the adjustable parameters and made them adjustable directly in the application when launching a job. These parameters include the main three and main two that affect the sensitivity and specificity for our two segmentation packages: NMS (Non-maximum suppression), probability, and pixel size for StarDist, and cell diameter and flow for CellPose. We have also included a markdown section in the application that explains the effects of these values when adjusted. We have defaulted the variables at values we optimized when performing MICSSS analysis and tested on the remaining technologies. See revised lines 194-199 for these changes.

While we acknowledge that adjusting other parameters in the pipeline may increase flexibility for an experienced user's specific use case, we fear that having an abundance of customizable parameters at the beginning of launching a job may confuse and overwhelm many users, especially without experience when analyzing these datasets. We champion the simplicity of our one pipeline for many use cases and technologies. In addition, changing the values of some variables or a combination of changes, while not breaking the pipeline, may incorrectly analyze the data and produce non-trustworthy results, which we hope to avoid. For example, for the values the reviewer mentions, we chose 3 microns, as it is about 30% of the cell area (assuming a diameter of 10 microns), which we observed is sufficient to account for any elastic registration error. If the value is increased much larger, we fear that adjacent cells will be considered the same cell and the user may be unable to segment cells correctly. Moreover, the NMS value in our StarDist algorithm essentially merges adjacent cells when increased, if the user intends to do this. Thus, to remove redundancy, promote simplicity for the user, and ensure the outputted data is scientifically accurate, we have chosen to not have the 3 micron parameter be adjustable. We have also altered this value to a percentage in line 157 of the text. As a next step, we are working on incorporating actual cell areas and thresholds when deciding to keep cells in the final composite segmentation mask. In regards to the 60% threshold, we again chose this from visual inspection when we analyzed panels with different numbers of stains, observing the benefits of the composite segmentation we display in Figure

2A. With only two stains, we wanted a cell to be identified if it was in both stains, but with about 10 stains, we observed that at least half of the stains had synonymous hematoxylin counter-stainings enough to recognize a consistent cell. We believe 60% to be a conservative threshold, in fact. For many of the same reasons and because we used 60% for the optimization, we have also hard-set this value and plan to make it adjustable in a future deployment of the code.

Comment 18

Also as mentioned in registration previously, this step is also dependent on the tiles and their size, so the similar arguments regarding the effect of the tile size in efficiency of the segmentation is prevailing here as well. Also the computational aspect of this step with regard to the number of the tiles would be relevant to discuss.

We thank the reviewer for this comment. The segmentation computational demands for a tile versus a whole-slide image was one of the reasons why we perform the segmentation on a per-tile basis. For more information on computational time and demands for registration on different tile sizes, see comment 12, and for specific steps in the pipeline, please refer to comment 50.

Comment 19

Quantification and Analysis:

The quantification is based on the artificial dilation of the nucleus mask. While this is quite mainstream, have authors considered the use of the cytoplasmic markers (in case they are present in the marker panel) to enhance their masks? While this is discussed in the discussion for future development, considering the importance of this issue, authors could consider already integrating an early version here.

We greatly appreciate this insightful comment, as it highlights a potential next step for enhancing our pipeline. Currently, the pipeline does not utilize cytoplasmic markers to define cytoplasmic boundaries but instead artificially expands nuclear boundaries to approximate them. While the pipeline is capable of analyzing cytoplasmic markers, the nuclear boundary expansion module remains unchanged for such analyses. Since our lab does not currently use cytoplasmic markers for MICSSS and the manuscript primarily focuses on validating MICSSS, we plan to include the ability to define cytoplasmic boundaries using true cytoplasmic markers in a future pipeline deployment.

Additionally, our unsupervised clustering has shown fair performance, and PCA analyses rarely indicate that outer pixel values (i.e., "artificial" cytoplasm and membrane) significantly drive clustering. Moving forward, we are concentrating on improving the nuclear segmentation model by integrating a completely new segmentation package (addressing comment 1) and comparing our StarDist composite model with the manual ground truth established by our pathologist (addressing comment 2).

Comment 20

Also, the two unsupervised methods used are appealing, and I suggest having a plot maybe in supplementary exhibiting the direct output of their use for an example input. Also, authors are discussing the unsupervised clustering of the cells and supervised clustering of the markers (L173) which is very confusing as I assume the markers intensity inevitably should be used for cell clustering and vice versa?

In general, the choice of K-means and GMM is rather unsupported in the text. One could consider multiple other methods such as NMF (for decomposing the sample and feature space simultaneously), Gamma mixture model, ICA, etc.

We thank the reviewer for this comment, as we have brainstormed many classification and clustering approaches and continue to do so for our next steps. We have added supplementary Figure S1, subpart C to depict the clustering runtimes for both mini batch K-means (MiniK) and UMAP-GMM approaches that we have tested and embedded into the pipeline. Although we explain the details and specific applications of either method in our methods section (see lines 478-504), we agree that this may not have been clear in the results section. We used mini batch k-means because of its speed and scalability for large datasets, and because these image data frames are massive arrays with gigabytes of information, we made it the default option. We hope this is made more clear with the added phrase we provided in lines 206-207. Also, we tested an additional method that the reviewer recommended (ICA) and tested four methodologies for our clustering step: 1) PCA-UMAP-GMM, 2) ICA-UMAP-GMM, 3) PCA-MiniK (our current methodology), and 4) ICA-MiniK. These results are depicted in Reviewer Figure 1. We evaluated different clustering methods using mutually exclusive markers (CD3, CD20, CD68) for eight whole-slide tumor samples and an error/overlapping plot for AEC intensity (0–1) to assess classification accuracy. Higher error rates, where a cell is assigned to multiple markers, were observed at lower intensities. Using an intensity threshold of 0.75 reduced the error to <1%, allowing us to classify a group of high-confidence positive cells. We can see consistently across the samples and markers that PCA-MiniK concentrates true positive cells in fewer clusters, while the other methods tend to spread them out over more clusters. Using this and the runtime information, we chose PCA-MiniK because it is computationally inexpensive and produces clusters with a higher proportion of true positive cells. In case of false positive cells, we deploy the re-clustering strategy on the fly. Our goal is not to classify the cells using PCA-MiniK; rather, it is to ease the burden on the pathologist who classifies them.

Comment 21

Minor:

The concept of the semi-supervised and automated are not clearly defined. Specially I think the semi-supervised is not a suitable term to be used here as this concept is commonly reserved in ML space for the cases where the dependent variable outcomes are not fully labeled. Considering the couple of times mentioning of 'machine learning' in the paper, it was a bit confusing to adopt the new concept the authors try to convey which in fact is the reclustering capability of cell types by users (user-mediated clustering, hybrid, semi-automotive?).

We thank the reviewer for this comment. We have revised, adapted and streamlined our terminology throughout the text. We agree that semi-supervised was incorrectly used to describe our method combining unsupervised clustering of cells with user-guided classification of marker expression patterns; therefore, we have replaced "semi-supervised" with "user-guided" throughout the text. Additionally, we have replaced "clustering" with "unsupervised clustering" any time our clustering methodology is introduced throughout the text. We also want to clarify that we use a third-party nuclear segmentation package that was pre-trained by the package authors. We have added "pre-trained" to any mention of this package throughout the text to clarify that we did not train this algorithm ourselves.

Comment 22

Reviewer Figure 1: Assessment of diverse clustering methodologies. Outputs of 1) PCA-UMAP-GMM, 2) ICA-UMAP-GMM, 3) PCA-MiniK (our current methodology), and 4) ICA-MiniK methodologies. Using a DICE threshold of 0.75 to compare the "true positive" cells deemed by our pathologist to the cells segmented by MARQO, we compared the True/False proportions across the clusters generated by each method for markers CD3, CD20, and CD68 for eight tumor samples.

The other one is the 'automated', for example mentioned on the L62-64 as desirable but again L80 refers to the issue arising from automated signal detection. I recommend authors either adopt alternative terms or be more clear about each of these and establish them as keywords for their specific use.

We appreciate that the reviewer pointed this out, as these incorrect terms may have misguided the reader. We refer the reviewer to comment 21, where we have adopted new language to describe the steps in our methodology. Specific to this comment, "automated" should now solely describe steps in the pipeline that do not require any user supervision or action, such as the unsupervised clustering of cell populations step.

Comment 23

I also think would be possible to talk more about the advanced ML models used in this space for example on cell detection [3-4] and in general providing more arguments supporting the development of this software as mentioned in L60-62.

We agree with the reviewer that many ML models exist in this space. While it's unfeasible to test and include every single one as they are released, we agree that using just the StarDist model, released when we began to develop the pipeline, may not be the most robust now, although still an accurate and high-performing cell segmentation mode that is widely used in the literature. Therefore, in combination with comment 1, we hope to have addressed this appropriately by adding a new segmentation model, CellPose. Please refer to our response to comment 1 for more information. We have added a thorough discussion of this pre-trained model to our methods section -- see lines 454-456. We have also added a more thorough discussion of AI tools and their future implementations in revised lines 323-329

Comment 24

L119, 368-370: It's great to have specially the affine registration for the alignment of the tiles and the program being able to forgo a registration step if it is not needed, however due to the computational cost of this transformation, does the program have automatic detection and bypass of this step in case this is not needed. And considering the parameters being optimized on MICSSS images, how transferrable the results would be to the other modalities. If they are tweakable, how users are made aware to choose the proper values for their respective modalities.

We thank the reviewer for this idea. Because the sequentially stained multiplex technologies we used will rarely align perfectly across stains, we did not apply a SSIM or similarity metric with a threshold to evaluate if we should perform the registration step. Rather, for multiplex technologies, we include the registration step and for non-multiplex technologies or multiplex technologies where the tissue does not undergo sequential staining (ex. CODEX), we skip the step. During registration per tile, we deconvolute the consistent nuclear counterstain and produce a vector field that would be used to register all color channels for the tile. Because this is a non-rigid registration, if the vector map has large magnitudes for many parts of the image (i.e. if there is significant tissue warping or damage), then the algorithm takes more computational power and time. If the vector map transforms the image minimally, then it takes much less computational power and time. Therefore, we are confident that tiles that do not require significant registration, although still undergoing the registration step, will not take significant time or resources to process. For the second point in this comment, we agree that the parameters should be freely accessible

to users with diverse modality applications, with the caveat that the user should have some familiarity in the imaging or computational space. We therefore refer the reviewer to comment 10, where we explain how we organized all the segmentation parameters in the application with corresponding descriptions of each. Because the pipeline's registration methodology is unique by consisting of a series of registrations and overlap with adjacent tiles to account for any tissue warping, we have hard-set these values. Despite this, they have been tested on multiple multiplex technologies and datasets. For parameters originating from third-party packages, we have appropriately cited that package and urge the user to visit that party's documentation. For registration-specific parameter inquiries, the user can visit the SimpleElastix GitHub.

Comment 25

L124 talks about the benchmarking data outputs generated. I think it would be great to have some of the most pivotal ones reflected in a graph as a supplementary figure.

We thank the reviewer for this comment. We have realized that the "benchmarking" outputs are specific to our clustering system (ex. timestamps, #CPUs used, etc.), and therefore we have revised this statement to reflect that the intermediate outputs are capable of being drag-and-dropped into other analysis platforms. like QuPath and Halo. Examples of these files are provided in Supplementary Table 1, as a graph would not display these types of large dataframes well. Please see revised lines 138-140 for this revision.

Comment 26

L128: It's suggestable to clearly explain the composite segmentation herein with reference to its underlying STARDIST core method upfront.

We agree with the reviewer to include the StarDist citation in both results and methods, and have thus adjusted our results text to include more details on the StarDist package with its citation. See lines 152-153 for this change.

Comment 27

L145: Can you provide your insight regarding the elimination of the red blood cells in terms of sensitivity and specificity rates?

We thank the reviewer for this comment. In parallel to selecting 15 new post-registered tiles outputted from the MARQO pipeline for our pathologist to manually quantify to assess and validate the composite segmentation performance, we additionally asked them to choose 5 of those 15 that comprised a large number of red blood cells (RBCs) so we could additionally address this comment. Because MARQO eliminates RBCs through a series of steps (composite segmentation, hematoxylin filtering, clustering, etc.), instead of reporting sensitivity/specificity for one step in the methodology, we rather assessed the composite elimination of RBCs when the pipeline outputs the final cell segmentation mask. We had our pathologist produce individual *.geojson* files by manually clicking on all RBCs in the 5 tiles. Then we compared these files to the masks produced by MARQO. RBCs selected manually but not kept in MARQO's final cell segmentation mask were correctly eliminated and considered "Hits" by MARQO's

RBC filtering algorithm. RBCs selected manually that did reside within any of the cell boundaries of the final cell segmentation mask produced by MARQO were considered "Misses." These results are organized into a contingency table in revised Supplementary Figure S3B and referenced in line 159. A quadrant of the tile that achieved the median performance out of the 5 tiles analyzed (99.5% Hit rate) is displayed in Supplementary Figure S3C. We hope these metrics and visuals help strengthen our argument for MARQO's ability to filter out RBCs in its final outputs.

Comment 28

L151: The semi-manual segmentation comparison seems a bit non-specific. In order to avoid confounding errors of two independent modes and gain more orthogonal insight, wouldn't it be more meaningful to have the comparison with fully automated segmentation and perfectly manual ones separately?

We agree with the reviewer that our previous comparisons of automatic segmentation using our composite StarDist model and semi-manual segmentation via QuPath were subpar. Thus, we refer the reviewer to comment 2, where we explain our new segmentation validation using a completely manually-counted ground truth by our pathologist.

Comment 29

L161: Why not comparing the outputs with at least one other mainstream method like Cellpose, specially so that the underlying method for MARQO is based on STARDIST?

We agree with the reviewer that many advanced segmentation models have recently been released, and while we want to strengthen our pipeline's ability to segment cells over time, we do not think it is the goal of this manuscript to compare the abilities of third-party segmentation tools. We have revised our validation method for the currently used composite StarDist model to include a completely manually-counted ground truth (see comment 2) and have also added an alternate, more recently-released segmentation module that the user may opt to use, named CellPose (see comment 1).

Comment 30

L174: You may consider using different terminology.

We thank the reviewer for this comment and refer them to comment 21.

Comment 31

L187-189: Is the manual counting of the positive cells performed on the raw image or the one after algorithmic cell detection is established?

We thank the reviewer for requesting this clarification. Manual counting was performed on raw images (or rather, smaller tiles) after registration but before segmentation. Using our new ground truth with manual

annotations method (see comment 2), our pathologist manually selected the X,Y coordinates in the raw image. We have adapted our methods section to reflect this work, explained in lines 623-667.

Comment 32

L214: What is the motivation for choosing these two markers?

We thank the reviewer for requesting added rationale as to why we chose these markers. To respond, we chose them because they are broadly used in the imaging space as T cell and tumor cell markers. Also the diversity of having a smaller cell specific membrane marker CD3 and a large cell specific cytoplasmic marker PanCK was advantageous when validating this module. We revised our text to include this rationale in lines 243-245.

Comment 33

L240-242: I recommend providing a preliminary figure in Supplement showing the result of your testing for each modality.

We thank the reviewer for this comment. We have added Supplementary Figure S7 to display other technologies (specifically Orion, CODEX, and CyclF) when being analyzed with MARQO in our visualization application.

Comment 34

L243: Referring to the high accuracy here is not quite convincing only based on the r-square and p-values. Why not considering the formal comparison of the MARQO performance with confusion matrix and its derived metrics such as F1 score?

We thank the reviewer for pointing this out. Although we wish we had the resources to perform manual ground truths for all technologies listed in Figure 5 and report the respective confusion matrices, we chose to validate only with our most-used multiplex technology MICSSS and defer to p-values and r-squares elsewhere.

Comment 35

L241: CyCIF

We thank the reviewer for recognizing this misspelling. We have revised all mentions of CyIF to CyCIF.

Comment 36

L268: 'tend 'to' be '

We thank the reviewer for recognizing this typo, which we have now fixed.

Comment 37

L438: It is unclear where these 60 GMMs are coming from!

We thank the reviewer for this comment. We empirically optimized the value of 60 GMMs by testing multiple different thresholds and observing how the clusters aligned to the correct respective marker types while also balancing computational time and resources. We added clarification in the text; kindly refer to lines 503-504.

Comment 38

Fig3: In general couple of points were not clear with these graphs that I think could be clarified. -how the random graphs are obtained? -what was the criteria for choosing the ROIs? -what are the dashed line in panel E?

We thank the reviewer for requesting more information about Figure 3. We have added a phrase to clarify each of the three questions in the figure legend, which can be found in lines 775-800 in the text.

Comment 39

Fig4: B: Where these boxes close up images are placed on the tissue?

We thank the reviewer for requesting this clarification. The boxes are indeed close-up blowup images of an ROI from the thumbnail sample or core adjacent to the blowups. We have revised the figure to depict where these ROIs originate from within the sample.

Comment 40

C: Are the points representing the tiles?

We thank the reviewer for this question. Each point represents an independently analyzed resected sample or TMA core. We have revised our Figure 4 legend to describe this better; see lines 802-814.

Comment 41

B: Typically the CD3 and PanCK are orthogonal, is that the reason why authors used these two here? Despite this fact, the right side of the 4th row is exhibiting the hump with a margin marked positive for both markers. How this is explained and in general, situations like these are calling for a more robust quantifications such as F1 score than linearity assessment emphasized throughout.

We thank the reviewer for probing at one of our next areas of improvement. You are correct that there may be some double-positive stained cells in the regions you describe in Figure 4B for CD3 and PanCK. Although we still champion the high accuracy rates determined for markers CD3 and PanCK across the myriad of tissue sizes and types, we agree that we could apply some scientific restraints to refine our classification technique and reduce false positive rates, especially for cells stained co-positive for two markers that are known not to usually be co-expressed together -- in this case, CD3 and PanCK. Our team is currently working on a post-processing module to do exactly this. While we tried not to promise too many next steps in the discussion section with future versions of the MARQO pipeline, we still have added two sentences to describe how we plan to tackle this issue. Please see lines 355-360 where we describe how we will try to reduce the number of falsely-determined doublet cells based on the frequencies of singlet positive cells, singlet negative cells, and proximity intensity values for the markers of interest.

Comment 42

Fig5: I recommend a more versatile comparison both in terms of markers and methods (other than QuPath), specially with reference to the ground truth for settled cell phenotypes. Like in panel B, PanCK, it seems there are at least two quite distinct cells to the left side of the image with positive signal that are not detected!

We thank the reviewer for this comment and kindly refer them to comments 1 and 34. While we optimized and validated both segmentation and classification modules with the MICSSS technology, we used a conventional QuPath-based methodology to compare other technologies. Because the composite StarDist segmentation performance may vary across technologies, we have also added a new segmentation model that the user can opt to use.

Comment 43

Fig6: A: What are the x-axis of this panel and are those meant to be vertically parallel? If they are the patients, why the one on the top is missing two and what is the order from the first heat map to the one below? B and C: Are these two lesions on the slide being treated differently? In general, depending on the analysis, the lesions on the same slide may (e.g. clearly in cases of neighborhood analysis) or may not (e.g. threshold gating) be treated independently. Would be good if authors could appreciate this issue specially regarding how MARQO will perform its analysis.

We thank the reviewer for helping us make Figure 6 clearer for the reader. For subfigure A, we have added an x-axis denoting patients and have also added gray boxes (with an updated legend) for patients who we could not acquire data for at baseline (ex. biopsy tissue was not acquired by the clinical team). Regarding subfigure B/C, while we agree that TMA cores can be treated as independent samples, we did not do the same for biopsied and resected tissues. Because of the nature of acquiring and processing the tissue, there were sometimes multiple fragments of tissue, or lesions, per sample. To streamline protocol and keep one sample per time point per patient consistent, we averaged together the metrics determined from both tissue lesions. This did not significantly affect our spatial analysis since we calculated the nearest neighbor for each pair of cell types. Because of the densities of the immune cells of interest, the nearest neighbor would almost always reside within the same lesion, regardless of where the adjacent lesion resided on the slide.

Comment 44

FigS1A: What would be the effect of having 'overlapping borders tiles removal' after application of vector field alignment?

We thank the reviewer for this question. During the registration process, each tile is expanded by 10% area and deconvoluted for its nuclear channel. This channel is used to determine the vector alignment field from the nth-stained tile to the first-stained tile, which is applied to all RGB channels. After this, the 10% overlap is removed from each tile. Because the tile was already registered from the vector field, there is no direct consequence of removing the 10% area on the vector field itself. The vector field is solely used to align the nth tile to the 1st stained respective tile. Kindly refer to comment 12 for a more detailed explanation of the registration process and how the 10% overlap is necessary.

Comment 45

FigS3: The quality of the image for FOXP3 is low and I was not able to discern the detections. However, for the PanCK, there are too many greens (and almost no blue) suggesting the high rate of false positive?

We thank the reviewer for pointing out these observations. When exporting the file to PDF, there must have been an error with the FoxP3 image. It is now fixed; kindly see Figure S5. Also, to better discern the detections, we have added the original zoomed-out tiles along with zoom-ins for each of the markers, which are zoomed in by about 75% compared to before this edit. Lastly, we agree that especially for PanCK, there were more green than blue annotations, indicating a high MARQO false positive rate. While PanCK remains a difficult marker to quantify due to its heterogeneous staining for diverse cell types, we believe this may have arisen because our pathologist was conservative in annotating true positive cells in high density areas.

Comment 46

SupTab1: CyCIF

We thank the reviewer for recognizing this typo, which we have now fixed.

Comment 47

SupTab2: Maybe removing the almost identical fourth sub-table by adding the PDL-1** to the end of the third sub-table as the 10th row, amending the title of the 3rd sub-table accordingly, and adding the note **: Only applicable to the MICSSS on whole-slide NSCLC resections.

We thank the reviewer for the suggestion to make our tables less redundant and our message more succinct. However, when performing MICSSS and replicating our work, the order of markers remains paramount, especially when staining different tissues. Moreover, while the NSCLC samples have PD-L1 staining, they also lack a MZB1 stain. For the complexities this may add to the table when combining, and because of the importance of marker order when staining lung versus liver tissue, we have kept the tables as they were.

Comment 48

The other overall issues that I believe are meaningful for the authors to consider is- the more versatile use of the markers as specially they are readily available throughout their study, specially immune ones like CD19/CD20 in last figure to detect possible community of the B cells suggestive of the tertiary lymphoid structures and maybe an additive figure as the pseudo-image of the exemplar tissue with the T can B cells overlaid.

We thank the reviewer for the idea to expound more on our immunological findings with an added visualization depicting the proximity of T and B cells. We have added images produced by the MARQO review application where we display CD3, CD8, and CD20 intensity values overlaid to the hematoxylin-stained tissue in both a high-density TLS-like region as well as a low immune cell density area. We also have performed localization analysis from the B cell perspective and taken note of their proximity to themselves and CD8 T cells in revised lines 304-307. We added the overlays and plots to supplementary Figure S8.

Comment 49

- discussing the optimal performance envelope of the software in general and specifically, with regard to patients' variability and best practices to diminish its adverse effect toward optimal take aways of the pipeline -something that is completely left out in the use case of the software reflected on the last figure.

We thank the reviewer for this comment. We agree that there lacked a discussion on the optimal performance envelope of the software, specifically in regard to patients' variability. Along with comments 10 and 24, we have added lines 333-334 to the discussion to conclude how the pipeline can be tailored to whatever use case the user sees fit by adjustment of the tweakable parameters. We also hope we have emphasized the benefit of having one streamlined pipeline that combines many analysis steps for a variety of inputs. In the discussion, we have added lines 332-333 to discuss the ability to simultaneously analyze multiple samples at once, despite patient variability. We hope this discussion better addresses the benefits of the pipeline.

Comment 50

- providing more arguments around the scalability and computational edges of this tool with regards to the big cohort studies possibly involving multiple imaging modalities and containing dozens to hundreds of samples.

We thank the reviewer for this comment. Our pipeline employs multi-node parallel computing techniques, combining multicore processing within nodes and distributed computing across multiple nodes to fully utilize all available hardware resources. For large cohort studies with diverse imaging modalities and hundreds of samples, the pipeline scales efficiently. With N tiles per sample and N cores distributed across nodes, the computation is fully parallelized, making the processing time per sample equivalent to the average time to analyze a single tile. This ensures high-throughput performance, with the only constraint being the available hardware resources. We have added a new supplementary Figure S1 to display average run times for a single tile throughout the steps in the pipeline -- for both MICSSS and mIF.

In addition, to reinforce our argument regarding MARQO's advanced multi-node distribution technique, we have added a total processing time versus number of cores available graph for a range of test case samples with an increasing number of tiles. However, because the pipeline is not fully automated and still requires an experienced user to launch the job and evaluate each cluster's positivity per marker, we agree that despite the scalability and computational efficiency of our pipeline, it still possesses the bottleneck of these user control steps. For example, for one sample stained via a typical MICSSS panel of 10 markers, we predict it would take the user approximately 5-10 minutes to launch the job, a variable amount of time for analysis depending on CPU availability, and 10-20 minutes total to classify all 15 clusters across all markers. Therefore, we will continue to try and automate quality control steps in future iterations, as appropriate, and have included the ability to quality control multiple samples at once, a novelty that will substantially reduce the amount of time taken by the user (i.e. 10-20 minutes to classify multiple samples). Please see revised lines 70-74 in the introduction and 127-131 in the results.

Comment 51

- placing their findings in the context of newly introduced and powerful transformer-based methods for deciphering the mesoscale architecture of the tissue [4-5].

We thank the reviewer for strengthening our text by including a more recent package that deciphers mesoscale architecture of whole-slide tissues. We agree that the step can be improved when our pathologist team produces manual tissue annotations in software such as QuPath. In a future deployment of the pipeline, we hope that we can embed a package that can do this more automatically, such as UNI, a general-purpose, self-supervised model for identifying tissue types in whole-slide images (reference 5 that the reviewer provided). Please see the added mention of how this package could be embedded into the MARQO pipeline in lines 409-411.

Comment 52

Lastly, as presented modules are exhibiting a natural ordinality, the authors may wish to consider, at their discretion, an acronym more reflective of this fact such as MIRSQA (Multiplex-imaging Integrative Registration Segmentation Quantification and Analysis) or alike!

We thank the reviewer for this suggestion. Because we are also pursuing a license and commercialization route and have used "MARQO" in our documentation there, we will keep the acronym. In addition, we want a name that is easy to enunciate and unique to competitive technology names. So far, we believe that "MARQO" fits those guidelines.

Bibliography

[1] Ma, J., Xie, R., Ayyadhury, S., Ge, C., Gupta, A., Gupta, R., ... & Wang, B. (2024). *The multimodality cell segmentation challenge: toward universal solutions. Nature methods*, 1-11.

[2] Spitzer, H., Berry, S., Donoghoe, M., Pelkmans, L., & Theis, F. J. (2023). *Learning consistent subcellular landmarks to quantify changes in multiplexed protein maps. Nature Methods*, 1-12.

[3] Liu, C. C., Greenwald, N. F., Kong, A., McCaffrey, E. F., Leow, K. X., Mrdjen, D., ... & Angelo, M. (2023). Robust phenotyping of highly multiplexed tissue imaging data using pixel-level clustering. *Nature Communications*, 14(1), 4618.

[4] Xu, H., Usuyama, N., Bagga, J., Zhang, S., Rao, R., Naumann, T., ... & Poon, H. (2024). A whole-slide foundation model for digital pathology from real-world data. *Nature*, 1-8.

[5] Chen, R. J., Ding, T., Lu, M. Y., Williamson, D. F., Jaume, G., Song, A. H., ... & Mahmood, F. (2024). Towards a general-purpose foundation model for computational pathology. *Nature Medicine*, 30(3), 850-862.

Remarks on code availability:

As the main offer of this study, I checked the GitHub repository, which is quite organized, with adequate instructions for installing the software with Docker. The README page is quite comprehensive with clear guidelines and step by step installation tips. The software interface was running smoothly without any particular issue. The data used in the manuscript is also provided. Running couple of examples, I see the conformity with the supplementary videos and obtaining outputs resembling the presented results in the paper.

I have not fully checked all the underlying code of the directories, but couple sampling and overall skimming suggested an organized and professional coding scheme throughout the full repository.

Reviewer #3 (Report for the authors (Required)):

This manuscript by Buckup et al describes an image alignment and analytic tool for multiplex immunohistochemical assays, which they term MARQO. The software is designed to handle a diversity of multiplex imaging approaches/platforms, including MICSSS. Ultimately all of the data input types are variation on theme - images of different targets obtained from the same sample - in some cases and image stack and in some cases, repetitive single-plex IHC.

Comment 53

The manuscript is clearly written, with well developed figures. The manuscript outlines the general capacities of the software package well. The authors compare the software to QuPath, Halo and Visiopharm, but do not reference other software packages that may be available - both commercial and research developed. The overall intended use of the software is analysis of a diversity of tissue inputs, including whole sections, biopsies and tissue microarrays with the analytic goal generally focused on immuno-oncology.

We sincerely thank the reviewer for their insightful comments. While our study used quantitative metrics to compare key components of our software to those in other platforms (e.g., QuPath, StarDist segmentation, Visiopharm), we did not perform exhaustive comparisons to all available software packages. Our pathologist team identified the QuPath workflow as the conventional standard in the field for these analyses. Therefore, we used it as our primary baseline for comparison, employing quantitative metrics to highlight the advantages of our innovative workflow. For other commercial and research-based software platforms, we focused on a qualitative discussion of their respective limitations relative to MARQO. We acknowledge that our comparison was selective, but it was deliberate to align with the most relevant methodologies. In response to the reviewer's suggestion, we have expanded the introduction to include additional technologies, such as ASHLAR and MCMICRO, for greater clarity. These updates can be found in line 78 of the revised introduction.

Comment 54

The workflow, image input and output parameters are well described and common to many image processing platforms. To describe the package in lacking in features would be unfair, however the software is limited in the tool set for which it can segment cells and offers no tools for extracellular matrix evaluation or other complex tools. The tools used for cell segmentation are reasonable, but are likely insufficient for some complex multi-cell type analytic settings.

We thank the reviewer for this comment and agree that it is unfair to claim the superiority of our platform without a thorough discussion of many of its limitations. We aimed to streamline and simplify multiplex image analysis for users and therefore created a classification module that binarizes cell positivity within the nucleus and an artificially expanded cytoplasmic area. In addition, our pipeline does not perform complex multi-cell type analysis, such as extracellular matrix evaluation. We have now added a mention of this limitation in line 354 of the discussion. With future deployments of the code, we hope to include more complex tools and abilities such as intensity scoring, extracellular evaluation, and membrane-specific segmentation.

Comment 55

The challenge is evaluation of the package. The primary argument the authors take is that MARQO is something that can be implemented, works on multiple multiplex datasets and is optimized for MICSSS. It is unclear how widely MICSSS is used. Although the authors provide some data comparing MARQO to the previously mentioned software package, they fail to demonstrate that MARQO has a clear benefit. The measure of benefit in image processing is generally performing an analysis that has not been possible previously, or demonstrating an outcome that can not be defined by other software, in other words superior results.

We thank the reviewer for this comment. While we aimed to showcase many of the benefits in image processing when using MARQO over other packages (one package for multiple steps, composite segmentation, user-guided classification, multiple built-in review modules in a downloadable GUI, project-level classification, etc.), we agree that we can strengthen our message and showcase more innovative features of the pipeline. To do this and address the reviewer's feedback, we have chosen to more closely showcase how our multi-step registration and composite segmentation process shows clear improvements to the currently-available methodology used in QuPath. We had our pathologist perform the analysis on QuPath with our TMA sample. They registered the same core biopsy across the six stainings using the affine registration feature. Then they used the StarDist segmentation tool in QuPath to identify cell nuclei in the first stain and applied that mask to the sequential stains, in order to visualize the performance of its registration better. The alignment of cell nuclei boundaries across stains depicts the performance and reliability of this methodology. In parallel, using MARQO, we extracted the intermediate files containing metadata for cell boundaries (.geojson) and the same corresponding tiles for the TMA analyzed in QuPath. We drag and dropped these files into QuPath to visualize the cell boundaries across stains in the same format that we did using the QuPath tools. From these parallel routes, we produced a figure depicting the alignment of cell boundaries for the first and last stain in the TMA panel. We have included a renal cell carcinoma and pancreatic duct carcinoma example in supplementary Figure S4 and have revised manuscript text lines 182-184. In addition, we have provided two more examples of breast cancer tissue in Reviewer Figure 2. From the visualizations, it can be seen that the QuPath methodology renders it impossible to accurately quantify cell positivity metrics by the last stain of the panel. We hope that these examples also portray the following messages: 1) MARQO is better than QuPath during elastic registration, 2) when a tissue moves during staining, MARQO will ignore it, and 3) when tissue is lost during staining, MARQO will ignore it. We hope these examples provide visual evidence that MARQO has clear benefits to the mentioned software packages when registering tissue and segmenting cells.

Comment 56

The software is good, but it fails to generate the enthusiasm to why a researcher would change pipelines if they are already analyzing data. The investigators discuss the benefits of local compute, however anyone doing these studies hopefully not limited by compute. Ultimately the real issue is total time - both human at computer and computer compute required to generate a usable data package. The measure of usability is a critical and commonly missing element of any evaluation of data pipelines, and as any experienced multiplex researcher, the process can take days to weeks to manage just one experiment.

We thank the reviewer for this comment. We hoped to generate enthusiasm for the user when listing the innovative items in paragraph two of the introduction: (1) using multiplex data to improve and refine our output data, (2) providing a downloadable GUI to make our pipeline accessible to both computational and non-computational users, and (3) combining an unsupervised clustering technique with a user-guided classification technique to augment closely-monitored but still efficient analysis by the user. However, we

Reviewer Figure 2: Registration and segmentation performance for QuPath versus MARQO. A) First stain (left) and last stain (right) region of interest for QuPath workflow (top) versus MARQO (bottom) for breast cancer stained by MICSSS with six total markers. **B)** First stain (left) and last stain (right) region of interest for QuPath workflow (top) versus MARQO (bottom) for breast cancer stained by MICSSS with six total markers. QuPath workflow includes nuclear segmentation on the first stain and applied to sequential stains, with sequential images registered with an affine transformation. The MARQO automated workflow includes composite segmentation and elastic registrations. Scale bars = 20 μ m.

agree that discussing the benefits of MARQO's advanced multi-node distribution architecture was lacking. To address this and respond to comment 50, we have provided a new supplementary Figure S1 providing more quantitative information on runtimes and computational resources when analyzing MICSSS and IF samples with variable number of tiles and markers. More specifically to address this comment and elicit enthusiasm from the user on the benefits of local and cluster computing, we have added a description of our unique architecture as an innovative item to paragraph two of the introduction. Please see lines 68-73 in the text. MARQO's architecture utilizes parallel computing and distributed systems/cloud computing, enabling workload distribution across multiple independent machines. This design allows investigators analyzing large cohorts of tens to hundreds of images, totaling thousands of tiles, to leverage on-demand cloud services or academic HPC environments. By scaling the number of machines, MARQO reduces analysis time per cohort to near the theoretical minimum — the average time required to process a single tile. In addition, our pipeline integrates a user-friendly GUI with unsupervised clustering and user-guided classification. The unsupervised clustering organizes cells into distinct groups, simplifying the process of positive cell identification. The intuitive interface broadens accessibility, allowing users with minimal training to participate in the analysis. By enabling a more distributed workload, the pipeline supports faster and more efficient experimental analysis across research teams. We hope that these edits and our responses here and to comment 50 help illustrate the improvements in usability when using MARQO, especially when analyzing larger samples that would usually take days to weeks to manage, and generate sufficient enthusiasm to change pipelines and use MARQO.

Remarks on code availability:

Although I like the idea of being able to review the code, many reviewers work in highly secure IT environments, and the capacity to download and review the code is not a viable approach from an IT security perspective.